# Cdo1-Camkk2-AMPK axis confers the protective effects of exercise against NAFLD in mice

Min Chen [1,2,3], Jie-Ying Zhu [1,2,3], Wang-Jing Mu [1,2,3], Hong-Yang Luo [1,2,3], Yang Li [1,2,3], Shan Li [1,2,3], Lin-Jing Yan [1,2,3], Ruo-Ying Li [1,2,3] & Liang Guo [1,2,3] ✉

Exercise is an effective non-pharmacological strategy for ameliorating non-alcoholic fatty liver disease (NAFLD), but the underlying mechanism needs further investigation. Cysteine dioxygenase type 1 (Cdo1) is a key enzyme for cysteine catabolism that is enriched in liver, whose role in NAFLD remains poorly understood. Here, we show that exercise induces the expression of hepatic Cdo1 via the cAMP/PKA/CREB signaling pathway. Hepatocyte-specific knockout of Cdo1 (Cdo1[LKO]) decreases basal metabolic rate of the mice and impairs the effect of exercise against NAFLD, whereas hepatocyte-specific overexpression of Cdo1 (Cdo1[LTG]) increases basal metabolic rate of the mice and synergizes with exercise to ameliorate NAFLD. Mechanistically, Cdo1 tethers Camkk2 to AMPK by interacting with both of them, thereby activating AMPK signaling. This promotes fatty acid oxidation and mitochondrial biogenesis in hepatocytes to attenuate hepatosteatosis. Therefore, by promoting hepatic Camkk2-AMPK signaling pathway, Cdo1 acts as an important downstream effector of exercise to combat against NAFLD.

Nonalcoholic fatty liver disease (NAFLD) is a progressive liver disease characterized by excessive intrahepatic fat storage (hepatic steatosis)[1]. Although it begins initially as asymptomatic fatty liver, NAFLD can progress to more severe diseases including non-alcoholic steatohepatitis, cirrhosis, and hepatocellular carcinoma[2–4]. Despite being an important target for the pharmacological industry, there are no approved pharmacotherapies for NAFLD[5,6]. Exercise is regarded as a currently effective non-pharmacological strategy for alleviating NAFLD[7], which promotes hepatic lipid catabolism, improves insulin resistance, and more importantly improves NAFLD-related metabolic disorders[8,9].

A randomized clinical trial of exercise-only trials in patients with NAFLD showed that structured exercise (vigorous moderate exercise, 150 minutes per week) reduced relative liver fat by 35% to 40% among patients with NAFLD[10]. A systematic review with meta-analysis of 12 pooled data from controlled adult human trials showed clear evidence for an apparent benefit of exercise against NAFLD[11]. A systematic review of 54 randomized controlled trials (RCTs) examined the impact of exercise intervention programs on selected cardiometabolic health indicators in adults with overweight or obesity. The results showed that intrahepatic fat decreased significantly after exercise interventions[12]. A meta-analysis incorporating data from 21 RCTs with a collective total of 1530 participants demonstrated that exercise interventions with a cumulative workload exceeding 10,000 kcal produced significant improvements in intrahepatic fat deposit[13]. A meta-analysis of 14 RCTs (551 subjects) revealed that exercise training subjects were more likely to achieve ≥30% relative reduction in magnetic resonance imaging (MRI)-measured liver fat than those in the control condition[14]. In a meta-analysis of 28 randomized trials, strong evidence was found that physical activity, independently from diet change, was associated

[1]School of Exercise and Health and Collaborative Innovation Center for Sports and Public Health, Shanghai University of Sport, Shanghai 200438, China. [2]Shanghai Frontiers Science Research Base of Exercise and Metabolic Health, Shanghai University of Sport, Shanghai 200438, China. [3]Key Laboratory of Exercise and Health Sciences of the Ministry of Education, Shanghai University of Sport, Shanghai 200438, China. ✉e-mail: guoliang@sus.edu.cn

with a significant reduction in intrahepatic lipid content and with reductions in alanine aminotransferase (ALT) and aspartate amino-transferase (AST)[15]. However, there is far less data available on how exercise impacts NAFLD, and the underlying mechanism of the posi-tive effects of exercise on the alleviation of NAFLD is still incompletely understood.

NAFLD is a primary metabolic disease state with an increasing prevalence rate but can be prevented with regular exercise[8,9,16,17]. Exercise has a powerful effect in the prevention and treatment of NAFLD, and it is even proved to be more effective than some low-performing pharmacological agents[18–21]. Regular exercise can improve NAFLD through both acute events and chronic adaptations driven by each exercise bout[7,10]. During exercise, a complex cellular and mole-cular network will be created and regulated, which contribute to the metabolic health effects of exercise[22–25]. Over the past years, a growing body of work has implicated hundreds or even thousands of molecules regulated by exercise that are involved in the regulation of whole-body metabolism[26–30]. Nevertheless, only a few of them had been char-acterized and had their biological functions in exercise been demonstrated[22,23,25]. Therefore, identification of the cellular and molecular signaling affected by exercise may lead to the development of therapeutic approaches that can, to some extent, mimic the effects of exercise for the prevention and treatment of NAFLD.

Cysteine dioxygenase type 1 (Cdo1) is a key enzyme for cysteine catabolism and the synthesis of taurine. Its expression is enriched in liver and adipose tissue, and is also found to be expressed in some other tissues[31]. Cdo1 is involved in a spectrum of physiological pro-cesses including adipogenesis[32], osteoblastic differentiation[33], redox homeostasis[34], fertility[35], bile acid metabolism[36], sulfide metabolism[35], etc. In pathophysiological processes, the methylation degree of Cdo1 promoter is associated with the progression and malignancy of tumors[11]. In addition, Cdo1 also improves metabolic disorders by taurine[37] and neurodegenerative diseases by regulating cysteine levels[38]. Notably, adipose tissue specific loss- and gain-of function experiment in mice showed that Cdo1 ameliorated diet-induced obe-sity and related metabolic disorders by promoting lipolysis[39]. These data indicate that Cdo1 is involved in a variety of physiological and pathophysiological processes. Yet, hitherto, few data have been available on the role of hepatic Cdo1 in NAFLD. Accumulating studies support the idea that fatty acid oxidation (FAO) and mitochondrial dysfunction are major players throughout the development of NAFLD[9,40,41]. Moreover, regular exercise exerts beneficial effects on improving mitochondrial function and FAO[9]. In this study, it was found that hepatic Cdo1 was induced by exercise. Hepatic Cdo1 tethers Camkk2 to AMPK by interacting with both of them, thereby activating AMPK signaling. This enhances mitochondrial biogenesis and FAO to inhibit hepatic steatosis. Our results demonstrate a pivotal role of hepatic Cdo1 in exercise-mediated improvement of NAFLD in mice.

## Results
### Hepatic Cdo1 is induced through the cAMP/PKA/CREB pathway during exercise in mice
To investigate the effect of exercise on Cdo1 expression, wild type C57BL/6 mice fed with chow diet (CD) were subjected to an established protocol of aerobic exercise (treadmill training) for 8 weeks (Fig. 1a), and it was found that the expression of Cdo1 in liver can be significantly upregulated after exercise (Fig. 1b, c). Cyclic AMP (cAMP), as an exer-cise response factor, can activate cyclic-AMP response element bind-ing protein (CREB) through protein kinase A (PKA) and thus promotes the transcription of target genes[42,43]. Consistent with the previous findings[43], exercise increased the content of cAMP and significantly increased CREB phosphorylation in liver (Fig. 1c–e). Because CREB is a transcription factor, the transcriptional role of CREB in Cdo1 expres-sion was investigated. Our chromatin immunoprecipitation (ChIP)

data revealed that CREB can bind to the promoter of Cdo1, and the binding to Cdo1 promoter is enhanced after treatment with Forskolin (Fig. 1f). Interestingly, a putative CREB binding sequence in the geno-mic regions of Cdo1 promoter was identified (Fig. 1g), indicating that transcription of Cdo1 may be directly regulated by CREB. Then, we constructed the Cdo1 promoter luciferase reporter gene and found that CREB enhanced the luciferase activity in a dose-dependent man-ner (Fig. 1h). When we mutated the site of Cdo1 promoter that could bind to CREB, the ability of CREB to enhance the luciferase activity was significantly weakened (Fig. 1g-i). Furthermore, Cdo1 mRNA and pro-tein levels were significantly up-regulated after Forskolin treatment in primary hepatocytes (Fig. 1j-l) and HepG2 cells (Fig. 1m-o). These results indicate that exercise can lead to the induction of hepatic Cdo1 through the cAMP/PKA/CREB pathway.

### Hepatocyte-specific knockout of Cdo1 (Cdo1LKO) impairs exercise-mediated alleviation of fatty liver in mice
To determine whether hepatic Cdo1 is potentially involved in NAFLD pathogenesis, an analysis of the published microarray data generated from the liver tissue of obese mice (GSE83596) and RNA sequencing (RNA-seq) data from the livers of patients (GSE126848) with or without NAFLD in the Gene Expression Omnibus (GEO) database was per-formed. Cdo1 expression was significantly lower in the livers of high-fat diet (HFD)-fed mice compared with chow diet (CD)-fed mice (Fig. 2a). Moreover, hepatic CDO1 was also significantly lower in NAFLD patients compared to non-NAFLD individuals (Fig. 2b). Furthermore, our data also showed that hepatic Cdo1 mRNA and protein in high-fat diet (HFD)-induced NAFLD mice were expressed at significant lower levels than those in control chow diet (CD)-fed mice (Supplementary Fig. 1a-g and Fig. 2c).

Our data encouraged us to investigate whether hepatic Cdo1 deficiency can affect exercise-mediated alleviation of fatty liver in mice. We generated hepatic-specific Cdo1 knockout (Cdo1LKO) mice and the control Cdo1flox/flox mice (WT control) (Supplementary Fig. 2a). The mice were fed with HFD for 16 weeks to construct NAFLD model. From the 9th week of HFD feeding, mice were subjected to an estab-lished protocol of exercise in rodents for 8 weeks (Fig. 2d). And incremental-load exhaustive exercise testing on a treadmill was con-ducted. As shown in Supplementary Fig. 2b, 8 weeks of exercise training led to significant increase of the running time to exhaustion in both WT and Cdo1LKO mice, which indicates that exercise training did enhance the exercise capacity of the mice. Compared with the non-exercise group of mice (Sed), treadmill exercise (Trained) significantly decreased body weight gain (Supplementary Fig. 2c, d), liver weight (Fig. 2e) of the mice. The WT/Trained mice livers had less lipid accu-mulation compared with WT/Sed mice, which was determined by H&E staining, Oil red O staining, triglycerides (TG) and cholesterol (TC) levels of mice liver (Fig. 2f–h). Meanwhile, compared with WT/Sed mice, the WT/Trained mice exhibited higher glucose tolerance and better insulin sensitivity (Fig. 2i–l), as well as lower serum level of alanine aminotransferase (ALT) and aspartate aminotransferase (AST) (Fig. 2m, n). However, Cdo1LKO led to more severe fatty liver pheno-types and blunted the role of exercise in the decrease of body weight gain (Supplementary Fig. 2c, d), and the alleviation of diet-induced hepatic steatosis and related metabolic disorders in mice (Fig. 2e–n). In addition, metabolic chamber measurements were performed in mice. Compared to their WT companions, Cdo1LKO mice had significant lower $O_2$ consumption rates, $CO_2$ emission levels and heat production (Supplementary Fig. 2e–g). However, Cdo1LKO did not affect food intake and physical activity of the mice (Supplementary Fig. 2h, i). Thus, decreased energy expenditure may contribute to the increased body weight gain which was mediated by Cdo1LKO. Because insulin sensitivity is affected by Cdo1LKO and exercise training, we further examined the insulin-stimulated Akt-Ser473 phosphorylation in HFD-fed mice liver. As shown, exercise enhanced insulin-induced Akt

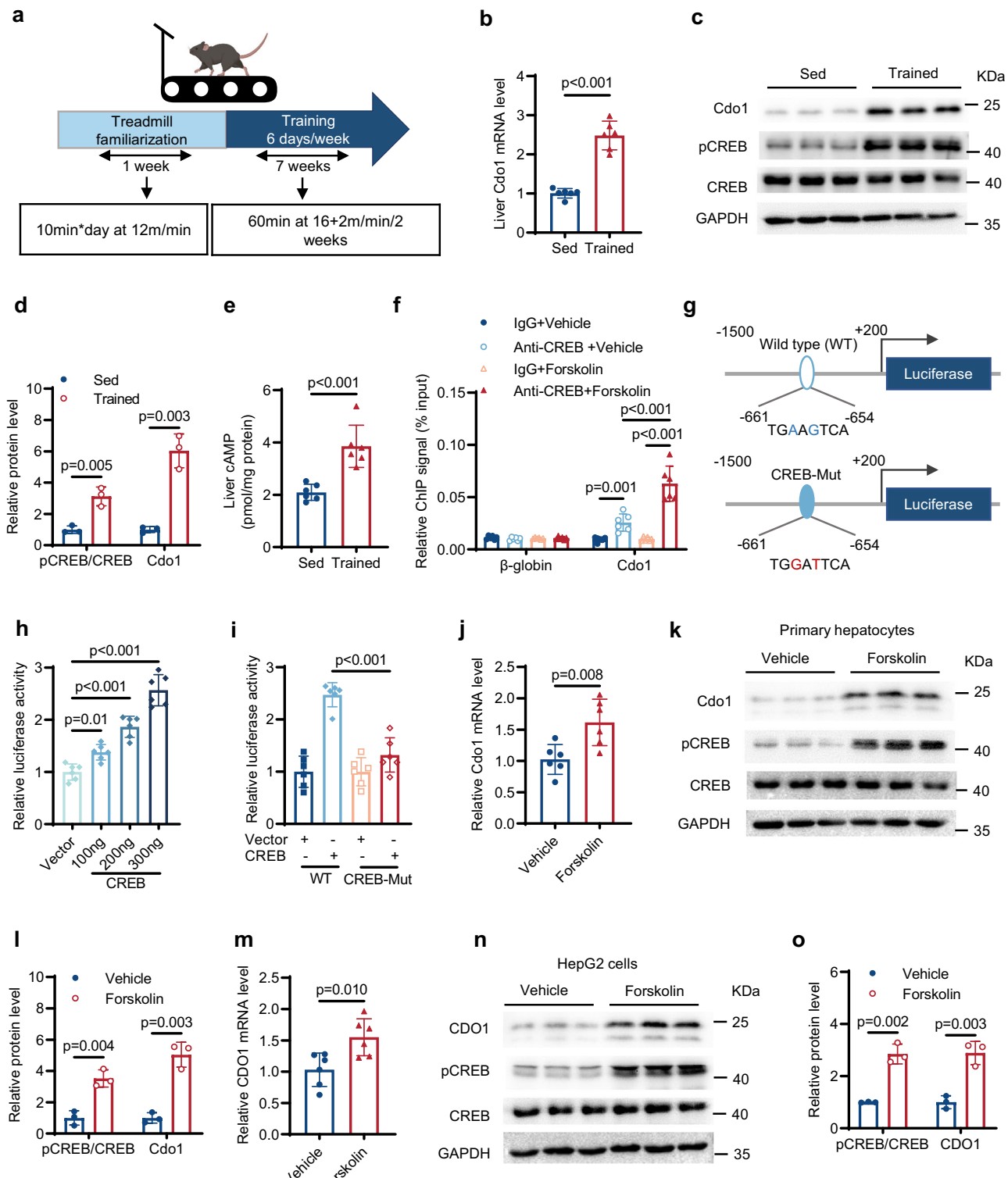

phosphorylation in the liver, which was attenuated by hepatic deficiency of Cdo1 (Supplementary Fig. 2j, k). Cdo1 can metabolize cysteine to generate taurine[31]. Therefore, we examined the levels of cysteine and taurine in the liver and found that neither exercise nor Cdo1[LKO] affected the levels of hepatic cysteine and taurine significantly (Supplementary Fig. 2l, m). We speculate that after the specific knockout of Cdo1 in the liver, Cdo1 existed in other tissues (such as fat, pancreas, etc.) may play a compensatory role in metabolizing cysteine. Taken together, these results above demonstrate an important role of hepatic Cdo1 in exercise-mediated alleviation of NAFLD in mice.

## Cdo1-regulated AMPK signaling is involved in exercise-mediated alleviation of fatty liver in mice

We next investigate the critical signaling pathways that are affected by Cdo1[LKO] in mice livers. We performed RNA sequencing (RNA-seq) to explore the differentially expressed genes (DEGs) in the wild-type and Cdo1[LKO] mice livers under the HFD-feeding condition. A total of 630 genes were significantly upregulated and 672 genes were notably downregulated in the samples of Cdo1[LKO] mice compared to control mice (Fig. 3a). The Gene ontology (GO) analysis revealed that DEGs were enriched in several biological processes, including positive

**Fig. 1 | Hepatic Cdo1 is induced through the cAMP/PKA/CREB pathway during exercise in mice. a** Treadmill training in chow diet-fed 8-week-old mice. The representing mouse model was created using BioRender.com. **b**–**e** Liver samples were isolated from the male mice after the exercise program shown in **a** is done. **b** The liver messenger RNA (mRNA) level of *Cdo1* ($n = 6$ mice per group). **c** Western blotting of mice liver lysates ($n = 3$ mice per group). **d** Quantification of western blotting results of **c**. **e** The cAMP level in mice liver ($n = 6$ mice per group). **f** CREB enrichment on the indicated gene promoters ($n = 6$ independent biological replicates). The β-globin gene promoter serves as a negative control. **g** Schematic representation of Cdo1 proximal promoter constructs used for luciferase assays. The predicted consensus of CREB-binding site is shown in the WT luciferase construct. The red letters indicate mutations of the CREB-binding site in the CREB-Mut construct. **h, i** Luciferase activities were measured in HEK293T cells. Data were normalized to the vector group ($n = 6$ independent biological replicates). **h** Cells were transiently transfected with WT (Wild type) reporter construct as shown in **g**, along with different amounts of CREB expression vector. **i** Cells were transiently transfected with WT or CREB-Mut reporter construct, along with control vector or CREB expression vector. **j**–**o** Primary mice hepatocytes or HepG2 cells were treated with 10 μM Forskolin for 12 h. **j** The mRNA level of *Cdo1* in primary hepatocytes ($n = 6$ independent biological replicates). **k** Western blotting of primary hepatocytes lysates ($n = 3$ independent biological replicates). **l** Quantification of western blotting results of **k**. **m** The *CDO1* mRNA level in HepG2 cells ($n = 6$ independent biological replicates). **n** Western blotting of HepG2 lysates ($n = 3$ independent biological replicates). **o** Quantification of western blotting results of **n**. Unpaired two-tailed t tests were performed in **b, d, e, j, l, m** and **o**; one-way analysis of variance plus Tukey's post hoc tests were performed in **h** and **i**; two-way analysis of variance plus Tukey's post hoc tests were performed in **f**. All data show the means ± SD. Source data are provided as a Source Data file.

regulation of lipid metabolism and mitochondrial respiratory chain complex assembly (Fig. 3b). This suggests that Cdo1 may regulate fatty acid metabolism as well as mitochondrial biosynthesis. Further studies indicated that the expression of some important FAO genes and mitochondrial biosynthesis genes was inhibited by Cdo1[LKO] in mice livers (Fig. 3c). And mitochondrial DNA copy number in mice livers was also decreased by Cdo1[LKO] (Fig. 3d). However, the genes related to lipogenesis was not affected by Cdo1[LKO] (Supplementary Fig. 2n). Furthermore, exercise could promote the expression of FAO genes and mitochondrial biosynthesis genes (Fig. 3c), and upregulate hepatic mitochondrial DNA copy number (Fig. 3d), but these effects were all blunted by Cdo1[LKO]. Similarly, the electron microscopic results showed that exercise could significantly increase the amount of mitochondria and improve the morphology of mitochondria in liver cells of control mice, while Cdo1[LKO] caused swelling and rupture of mitochondria, and exercise could not improve the ultrastructure of mitochondria in the liver cells of Cdo1[LKO] mice (Fig. 3e).

The Kyoto Encyclopedia of Genes and Genomes (KEGG) pathway analysis revealed that these upregulated genes (Cdo1[LKO] vs control WT) were enriched in a variety of pathways, including nuclear factor kappa-B (NF-κB) signaling pathway and C-type lectin receptor signaling pathway (Supplementary Fig. 2o), whose roles in Cdo1-regulated hepatic homeostasis of lipid metabolism merits further investigation. And the downregulated genes were significantly enriched in multiple pathways, including AMP-activated protein kinase (AMPK) pathways (Fig. 3f). Next, we examined the signal of AMPK in mice livers. The western blotting results showed that the phosphorylated (p)-AMPKα (Thr172) level increased and the level of phosphorylated acetyl CoA carboxylase 1 (p-ACC1) was also increased by exercise, but those effect were abolished by Cdo1[LKO] (Fig. 3g and Supplementary Fig. 2p). It is known that AMPK can promote the expression of mitochondrial bio-synthesis and FAO genes, and thus alleviate hepatocyte steatosis[44,45]. Furthermore, the AMPK signaling pathway is thought to be critical in the alleviation of NAFLD by exercise[31,46]. Thus, hepatic Cdo1 could promote FAO and mitochondrial biogenesis to inhibit hepatic steatosis, in which AMPK signaling could be involved.

## Cdo1 deficiency exacerbates hepatocytes steatosis in vitro

We further evaluated the role of Cdo1 in hepatic steatosis by treating primary hepatocytes from Cdo1[flox/flox] mice with oleic acid (OA) and palmitic acid (PA). In primary hepatocytes, the accumulation of lipid droplets, as observed by Nile Red and TG assay, increased significantly after the intervention of OA and PA (Supplementary Fig. 3a, b). The primary hepatocytes were treated with adenovirus harboring Cre (AD-Cre) to knock out Cdo1 or adenovirus harboring β-galactosidase gene z (AD-LacZ) as a control. Compared to control hepatocytes, lipid accumulation was significantly increased in Cdo1 knockout (KO) hepatocytes after OA and PA treatment (Fig. 4a and Supplementary Fig. 3c). Based on the results in Fig. 3, we hypothesized that impaired AMPK

signaling could be involved in Cdo1 deficiency-induced hepatic steatosis. Therefore, we measured AMPK activity and its downstream factor ACC1. Compared to control hepatocytes, p-AMPKα and p-ACC1 protein levels were reduced in Cdo1 KO hepatocytes (Fig. 4b and Supplementary Fig. 3d). AMPK signaling is known to promote FAO and mitochondrial biogenesis to protect against hepatosteatosis. Consistently, expression of genes encoding FAO and mitochondrial bio-genesis, including *Pparα, Cpt1a, Ppargc1a, mt-Co1*, were down-regulated in Cdo1 KO hepatocytes (Fig. 4c). In line with this, Cdo1 KO substantially decreased mitochondrial biogenesis (Fig. 4d) and the copy number of mitochondrial DNA (Fig. 4e) in primary hepatocytes. Next, we explored the effects of Cdo1 KO on mitochondrial function. Notably, Cdo1 KO cells exhibited a significantly lower capacity for basal respiration and maximal respiration (Fig. 4f), indicating that Cdo1 KO resulted in lower mitochondrial activity. In addition, the mitochondrial FAO rate was also decreased (Fig. 4g). Moreover, we used small interfering RNA (siRNA) transfection to knock down CDO1 in HepG2 cells and similar results were obtained (Fig. 4h-n and Supplementary Fig. 3e-h). These in vitro data indicate that Cdo1 deficiency in hepatocytes decreases mitochondrial biogenesis and FAO, thereby exacerbating hepatosteatosis.

## Cdo1 overexpression alleviates hepatocytes steatosis in vitro

Then, gain-of-function experiments were performed to investigate whether Cdo1 overexpression could attenuate hepatic steatosis in vitro. We extracted primary hepatocytes from WT mice, which were then infected with adenovirus for overexpression of wild-type (Cdo1[WT]) or Cdo1 mutant (Cdo1[Y157F]). The Cdo1[Y157F] has been shown to exhibit little enzymatic activity[39]. As expected, overexpression of Cdo1[WT] alleviated hepatocyte steatosis (Figs. 5a, b). And over-expression of Cdo1[WT] enhanced AMPK signaling (Fig. 5c, d), the expression of FAO and mitochondrial biogenesis genes (Fig. 5e), the copy number of mitochondrial DNA (Fig. 5f), mitochondrial bio-genesis (Fig. 5g), basal respiration and maximal respiration (Fig. 5h) and FAO level (Fig. 5i). Interestingly, hepatocytes steatosis, AMPK signaling activation, mitochondrial function and FAO level were also improved after overexpression of Cdo1[Y157F] (Fig. 5a–i). Furthermore, similar results were also obtained when overexpressing CDO1[WT] or CDO1[Y157F] in free fatty acids (FFAs)-treated HepG2 cells (Supplementary Fig. 4a-i). These results indicate that Cdo1, as well as its enzyme activity deficient mutant, can activate AMPK signaling to promote mitochondrial biogenesis and FAO, thereby ameliorating hepatocytes steatosis.

## AMPK signaling is indispensable for Cdo1-mediated amelioration of hepatocytes steatosis

Our animal experiment reveals that exercise can promote AMPK signaling in mice liver, and the activity of AMPK signaling in the liver of Cdo1[LKO] mice is significantly lower than control WT mice. Moreover,

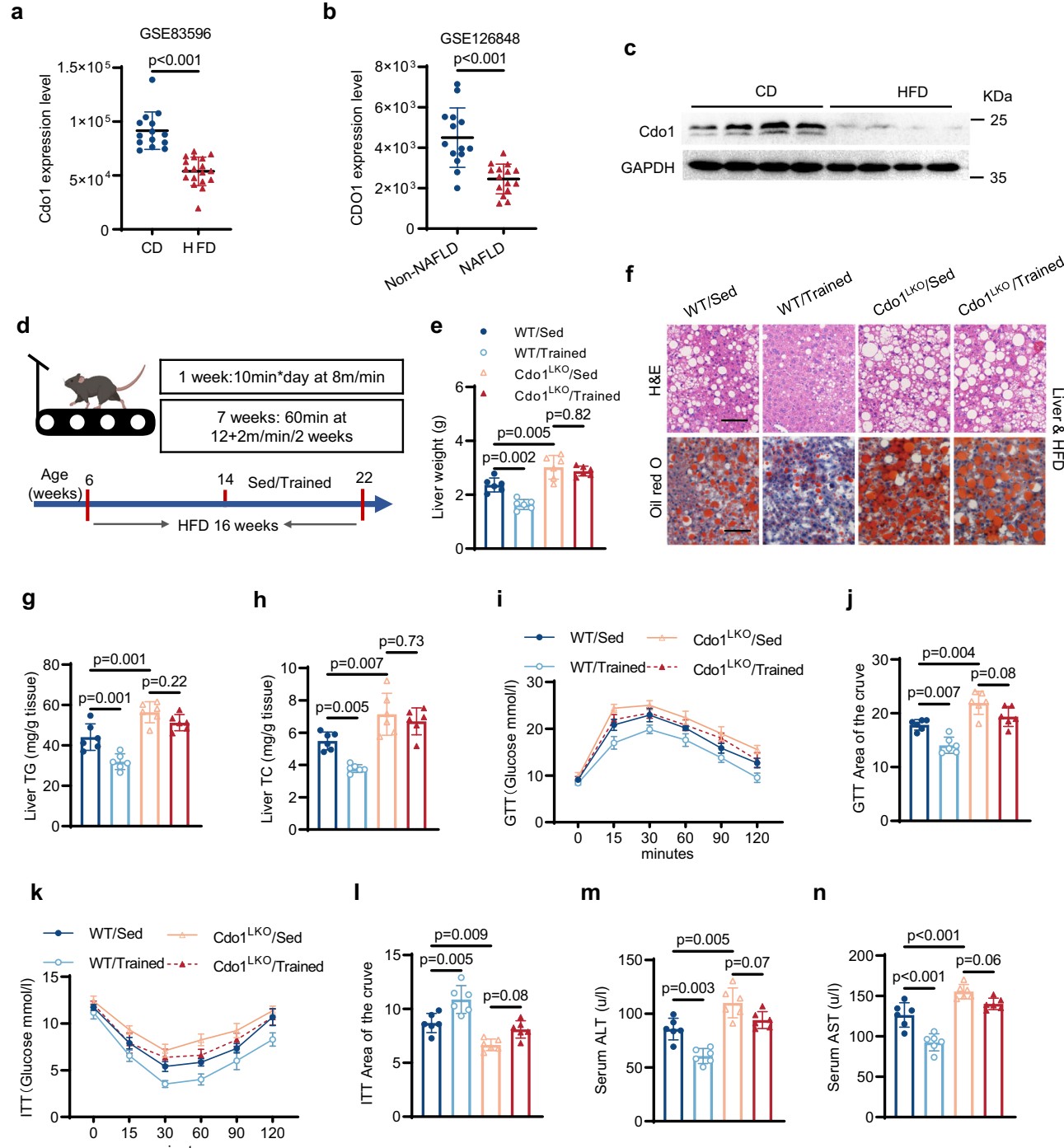

**Fig. 2 | Hepatocyte-specific knockout of Cdo1 (Cdo1$^{LKO}$) impairs exercise-mediated alleviation of fatty liver in mice. a** Scatter diagram indicating the relative *Cdo1* gene expression level in the high-fat diet (HFD)-fed mice group relative to the normal chow diet (CD)-fed controls from the GEO database in a microarray assay ($n = 14$ mice in CD group, and $n = 18$ mice in HFD group). **b** Scatter diagram indicating the relative *CDO1* gene expression level in patients with non-NAFLD people and NAFLD patients from the GEO database in high-throughput RNA sequencing ($n = 14$ individuals in non-NAFLD group, and $n = 15$ individuals in NAFLD group). **c** 6-week-old male mice were fed CD or HFD for 16 weeks before being sacrificed. The mice liver lysates were analyzed by western blotting ($n = 4$ mice per group). **d** 6-week-old Cdo1$^{flox/flox}$ (WT) and Cdo1$^{LKO}$ male mice were fed HFD for 16 weeks, with or without exercise in the last 8 weeks. The intervention program is illustrated. The representing mouse model was created using BioRender.com.

**e–n** Mice were treated as described in **d** before being sacrificed for analysis ($n = 6$ mice per group). **e** Mice liver weights. **f** Representative images of hematoxylin and eosin (H&E) staining and Oil Red O staining of liver sections. Experiments were performed 3 times and similar results were obtained. Scale bars, 50 μm.
**g, h** Triglyceride (TG) and cholesterol (TC) levels in mice livers, respectively.
**i** Glucose tolerance test (GTT) was performed in mice under 14 weeks of HFD feeding. **j** Analysis of the GTT data in **i**, with subtraction of the basal glucose to generate an area of the curve (AOC). **k** Insulin tolerance test (ITT) was performed in mice fed with HFD for 15 weeks. **l** Analysis of the ITT data in **k** with AOC. **m, n** Serum alanine aminotransferase (ALT) and serum aspartate aminotransferase (AST) levels in mice, respectively. Unpaired two-tailed t tests were performed in **a** and **b**; two-way analysis of variance plus Tukey's post hoc tests were performed in **e, g, h, j** and **l–n**. All data show the means ± SD. Source data are provided as a Source Data file.

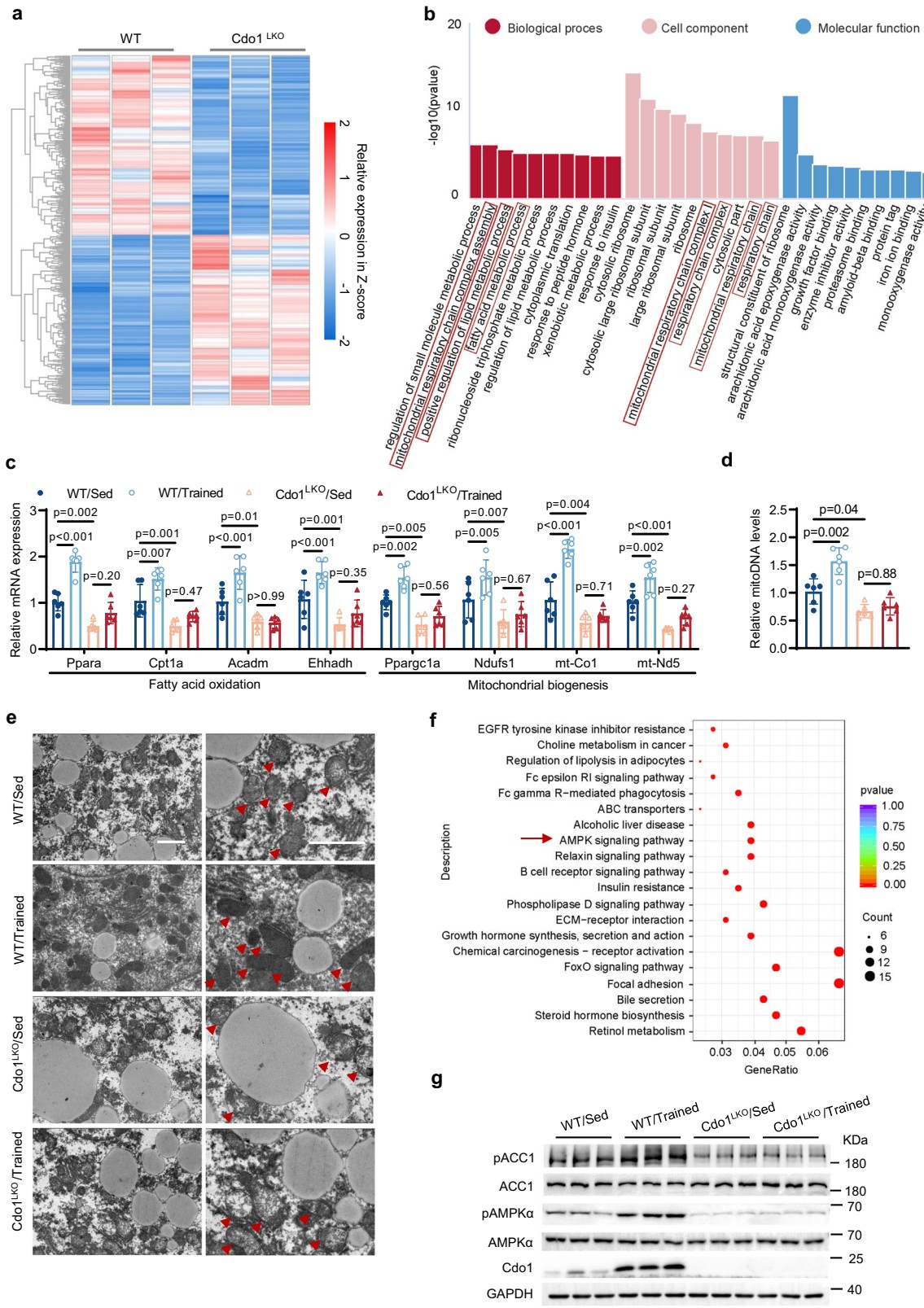

**c** (Fatty acid oxidation | Mitochondrial biogenesis)

**e** WT/Sed, WT/Trained, Cdo1^LKO/Sed, Cdo1^LKO/Trained

overexpression of Cdo1 and its mutant promotes AMPK signaling in hepatocytes. These results prompt us to explore the potential role of AMPK signaling in the Cdo1-mediated amelioration of hepatocytes steatosis. When we used short hairpin RNA (shRNA) to knock down AMPKα in primary hepatocytes, it was found that the effects of over-expressing Cdo1^WT or Cdo1^Y157F on the alleviation of hepatic steatosis were abolished (Fig. 6a, b). Consistently, after the knockdown of

AMPKα, Cdo1^WT or Cdo1^Y157F-mediated upregulation of FAO and mitochondrial biogenesis genes expression, decrease of p-ACC1, enhancement of mitochondrial biogenesis and respiration, and FAO were significantly attenuated (Fig. 6c–i). Similar results were obtained in HepG2 cells (Supplementary Fig. 5a–i). Collectively, these data indicate that the activation of AMPK signaling is required for Cdo1-mediated amelioration of hepatosteatosis.

**Fig. 3 | Cdo1-regulated AMPK signaling is involved in exercise-mediated alleviation of fatty liver in mice. a** Heatmaps representing the upregulated and downregulated genes (Cdo1[LKO] vs WT) in the livers of HFD (High-fat diet)-fed mice without exercise which was obtained from the RNA sequencing (RNA-seq) data ($n = 3$ mice per group, $p < 0.05$). **b** Gene ontology (GO) pathway analysis based on the RNA-seq data as described in **a** ($p < 0.05$). **c**–**g** Mice were treated as described in Fig. 2d. **c** The mRNA levels of the indicated genes were determined in mice livers ($n = 6$ mice per group). **d** The mitochondrial DNA (mitoDNA) levels in mice livers ($n = 6$ mice per group). **e** Transmission electron microscopic analysis (HITACHI HT7800) of mice livers. Scale bar: 2μm. Arrowheads indicate the mitochondria. Experiments were performed 3 times and similar results were obtained. **f** Kyoto Encyclopedia of Genes and Genomes (KEGG) pathway analysis of downregulated genes based on RNA-seq data in **a** ($p < 0.05$). "Count" means the number of genes that are in the corresponding KEGG pathway. **g** The mice liver lysates were analyzed by western blotting ($n = 3$ mice per group). Two-sided Wald tests without adjustment were performed in **a**, **b** and **f**. Two-way analysis of variance plus Tukey's post hoc tests were performed in **c** and **d**. All data show the means ± SD. Source data are provided as a Source Data file.

## Cdo1 tethers Camkk2 to AMPK by interacting with both of them, thereby promoting AMPK phosphorylation

The mechanism of how Cdo1 promotes AMPK signaling was then investigated. We infected the primary hepatocytes with adenoviruses harboring the gene encoding Flag-tagged Cdo1 (AD-Flag-Cdo1) and subjected the cell lysates to co-immunoprecipitation (CoIP) by using anti-Flag antibody, followed by western blotting (Fig. 7a). LKB1 and Camkk2 are two major upstream kinase to phosphorylate and activate AMPKα[47,48]. Pulldown assays showed that Cdo1 could interact with both Camkk2 and AMPKα, but not LKB1 (Fig. 7a and Supplementary Fig. 6a, b). Similarly, we found that the Cdo1[Y157F] could also interact with AMPKα and Camkk2 in primary hepatocytes (Supplementary Fig. 6c–e). In consistence with the pulldown experiments, confocal analyses showed that Flag-Cdo1[WT] and Flag-Cdo1[Y157F] could co-localise with HA-Camkk2 in Hepa1-6 and HEK293T cells (Fig. 7b, c and Supplementary Fig. 6f, g, respectively). Furthermore, at the endogenous level, Camkk2 could also interact with Cdo1, as well as with AMPKα (Fig. 7d and Supplementary Fig. 6h, i). These data indicate that Cdo1, Camkk2, and AMPKα could coexist in the same complex.

We then investigate whether Cdo1 could facilitate the association of AMPKα and Camkk2 in primary hepatocytes. It was found that overexpression of Cdo1[WT] or Cdo1[Y157F] promoted the interaction between AMPKα and Camkk2 (Fig. 7e, f and Supplementary Fig. 6j), and knockout of Cdo1 significantly weakened the interaction between AMPKα and Camkk2 (Fig. 7g, h and Supplementary Fig. 6k). Furthermore, in vitro cell-free reconstitution experiment was also performed by using recombinant Cdo1, Camkk2 and AMPK proteins. In this in vitro cell-free kinase assay, it was found that Cdo1 also promoted Camkk2-induced Thr172 phosphorylation of AMPKα (Fig. 7i, j and Supplementary Fig. 6l), which is consistent with the cellular experiments for testing the role of Cdo1 in regulating AMPK kinase activity (Fig. 5c and Supplementary Fig. 4c). To clarify the role of LKB1 and Camkk2 in the effects being observed, experiments were performed in mice primary hepatocytes with LKB1 knockdown or Camkk2 knockdown. As shown in Fig. 7k, l and Supplementary Fig. 6m, knockdown of Camkk2 in primary hepatocytes impaired Cdo1 overexpression-mediated activation of AMPK phosphorylation on Thr172. In contrast, the knockdown of LKB1 had little effect on Cdo1 overexpression-mediated activation of AMPK phosphorylation on Thr172 (Fig. 7m, n and Supplementary Fig. 6n). These data indicate that Camkk2, but not LKB1, is required for the role of Cdo1 in promoting AMPK activation. These results indicate that Cdo1 can promote the association of Camkk2 with AMPK by interacting with both of them, thereby facilitating Camkk2-mediated phosphorylation of AMPK.

## Hepatocyte-specific overexpression of Cdo1 (Cdo1LTG) and exercise synergistically alleviate NAFLD in mice

We next carried out gain-of-function experiments by using the albumin promoter-triggered Cre expression to specifically drive the expression of 3*Flag-tagged Cdo1 in liver tissue for the generation of hepatic-specific Cdo1 transgenic (Cdo1[LTG]) mice (Supplementary Fig. 7a). The 6-week-old Cdo1[LTG] and control WT mice were fed with HFD for 16 weeks and were subjected to treadmill training or not in the last 8 weeks (Supplementary Fig. 7b). Incremental-load exhaustive exercise testing on a treadmill was conducted in both WT and Cdo1[LTG] mice after 8 weeks of exercise training (Supplementary Fig. 7c). Compared to the control group (WT/Sed), treadmill exercise (WT/Trained) or hepatic overexpression of Cdo1 (Cdo1[LTG]/Sed) decreased the body weight gain (Supplementary Fig. 7d, e), liver weight in mice (Supplementary Fig. 7f). As a result of the effect of CDO1[LTG] on body weight gain in mice, we performed metabolic chamber measurements on mice. Compared to their WT companions, Cdo1[LTG] mice had significant higher $O_2$ consumption rates, $CO_2$ emission levels and heat production (Supplementary Fig. 7g-i). However, Cdo1[LTG] did not affect food intake and physical activity of the mice (Supplementary Fig. 7j, k). Thus, increased energy expenditure may contribute to the reduced body weight gain which was mediated by Cdo1[LTG]. In addition, exercise alone or Cdo1[LTG] alone also decreased the lipid accumulation in the liver (Fig. 8a–c), improved glucose tolerance and insulin sensitivity (Fig. 8d-g), promoted insulin-induced Akt-Ser473 phosphorylation (Supplementary Fig. 7l, m), led to lower serum level of ALT and AST (Fig. 8h, i), promotes AMPK activity (Fig. 8j and Supplementary Fig. 7n), upregulated the expression of FAO and mitochondrial biogenesis genes (Fig. 8k), and increased mitochondrial DNA copy number in mice livers (Fig. 8l). It is worthy of mentioning that the combination of exercise and Cdo1[LTG] (Cdo1[LTG]/Trained) yielded better effects on inhibiting fatty liver, promoting glucose tolerance and insulin sensitivity, increasing insulin-induced Akt phosphorylation, reducing liver injury, enhancing FAO and mitochondrial biogenesis (Fig. 8a-l and Supplementary Fig. 7l–n). Hepatic taurine and cysteine levels were not affected significantly by exercise, Cdo1[LTG] or their combination (Supplementary Fig. 7o, p). These results demonstrated that Cdo1[LTG] and exercise can cooperate to alleviate NAFLD in mice.

The experiments above were performed in male mice. To examine whether these findings above also apply to female mice, the role of hepatic Cdo1 in exercise-mediated alleviation of NAFLD in female mice was investigated. As shown in Supplementary Fig. 8a-g, hepatocyte-specific knockout of Cdo1 in female mice impaired exercise-mediated reduction of liver weight, liver TG/TC levels, serum ALT/AST levels, and attenuated exercise-mediated upregulation of FAO genes and mitochondrial biosynthesis genes expression. On the contrary, hepatocyte-specific overexpression of Cdo1 in female mice ameliorated NAFLD in synergy with exercise (Supplementary Fig. 9a-g). Therefore, our data indicates that hepatic Cdo1 is an important effector of exercise to ameliorate NAFLD both in male and female mice.

## Discussion

In this study, we found that exercise could promote the expression of hepatic Cdo1 through cAMP/PKA/CREB signaling pathway (Fig. 9). Cdo1[LKO] impairs exercise-mediated decrease of lipid accumulation, improvement of mitochondrial biogenesis and FAO in hepatocytes, thereby disrupting exercise-mediated alleviation of fatty liver in mice. Cdo1[LTG] and exercise can cooperate to promote mitochondrial biogenesis and FAO, which synergistically ameliorate hepatic steatosis in mice. Mechanistically, the Cdo1-Camkk2-AMPK axis in hepatocytes confers the protective effects of exercise against NAFLD (Fig. 9). We found that hepatic Cdo1 is an exercise-responsive factor to help to combat NAFLD.

Exercise is recognized as a non-pharmacological strategy to alleviate the pathology of NAFLD. However, our understanding of the

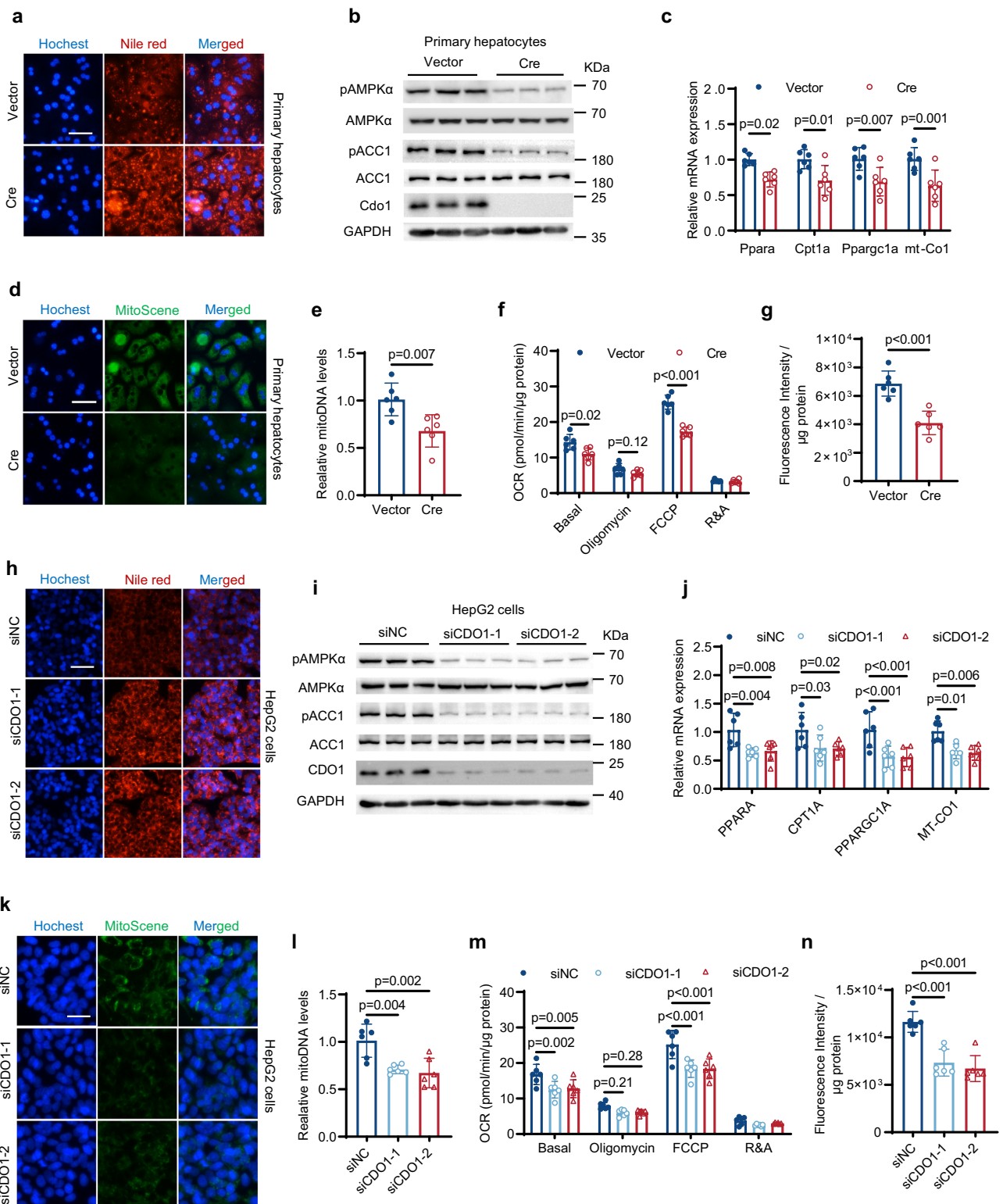

mechanisms by which exercise induces molecular changes in liver tissue is limited[8]. Cdo1 is an enzyme for the cysteine catabolism and taurine production, which is highly expressed in liver tissue[31]. The analysis of the GEO database showed that the expression of CDO1 in the liver of NAFLD patients was significantly decreased compared to non-NAFLD individuals (Fig. 2b). However, the specific function of liver Cdo1 in NAFLD is rarely reported. In our loss- and gain-of-function animal study, we demonstrated that the liver-specific knockout of Cdo1 (Cdo1$^{LKO}$) led to more severe fatty liver (Fig. 2f-h). Moreover,

Cdo1$^{LKO}$ impaired exercise-mediated positive effects against NAFLD (Fig. 2). However, liver-specific overexpression of Cdo1 (Cdo1$^{LTG}$) can ameliorate NAFLD in mice (Fig. 8). Importantly, Cdo1$^{LTG}$ and exercise cooperate to alleviate hepatocyte steatosis. Our in vitro experiments verified that Cdo1 can alleviate hepatocyte steatosis by promoting mitochondrial biogenesis and FAO. Some previous studies have also shown that Cdo1 is involved in promoting lipid metabolism and mitochondrial function. Our previous research showed that adipose-specific knockout of Cdo1 in mice impairs energy expenditure, cold

**Fig. 4 | Cdo1 deficiency exacerbates hepatocytes steatosis in vitro.** From **a**–**g** Cdo1^flox/flox loci-containing primary hepatocytes were infected with adenovirus harboring Cre (AD-Cre) for the ablation of Cdo1. After 24 h, cells were treated with a mixture of 0.6 mM FFAs (oleate and palmitate) at a final ratio of 2:1 for 24 h. Then cells were harvested and analyzed. $n = 3$ independent biological replicates for western blotting. $n = 6$ independent biological replicates for other experiments. **a** Nile red and Hoechst staining of the cells. Scale bars, 50 μm. **b** Western blotting of the cell lysates. **c** The mRNA levels of the indicated genes. **d** MitoScene and Hoechst staining of the cells. Scale bars, 50 μm. **e** The mitochondrial DNA (mitoDNA) levels of the cells. **f** Oxygen consumption rate (OCR) of the cells. FCCP, mitochondrial uncoupler. R&A, rotenone and antimycin A. **g** Fatty acid oxidation (FAO) levels of the cells. **h**–**n** HepG2 cells were transfected with the indicated small interfering RNA (siRNA). After 24 h, cells were treated with a mixture of FFAs as described above for 24 h. Then cells were harvested and analyzed. $n = 3$ independent biological replicates for western blotting. $n = 6$ independent biological replicates for other experiments. **h** Nile red and Hoechst staining of the cells. Scale bars, 50 μm. **i** Western blotting of the cell lysates. **j** The mRNA levels of the indicated genes. **k** MitoScene and Hoechst staining of the cells. Scale bars, 25 μm. **l** The mitoDNA levels in HepG2 cells. **m** OCR in HepG2 cells. **n** Fatty acid oxidation (FAO) levels in HepG2 cells. Experiments were performed 3 times and similar results were obtained in **a, b, d, h, i** and **k**. Unpaired two-tailed t tests were performed in **c, e** and **g**; one-way analysis of variance plus Tukey's post hoc tests were performed in **j, l** and **n**; two-way analysis of variance plus Tukey's post hoc tests were performed in **f** and **m**. All data show the means ± SD. Source data are provided as a Source Data file.

tolerance and lipolysis, exacerbates diet-induced obesity and decreases adipose expression of the key lipolytic genes. However, mice with transgenic overexpression of Cdo1 leads to the opposite results[39]. Another study also showed that knockdown of *CDO1* gene in adipose-derived mesenchymal stem cells (ASC52telo cells) and human adipocyte precursor cells led to decreased mitochondrial respiratory function (basal and maximal respiration and oxygen consumption for ATP production)[49]. These results suggest that Cdo1 may play an extensive role in promoting fatty acid catabolism and mitochondrial function. Next, we investigate the critical signal pathway underlying the protective role of Cdo1 against hepatosteatosis. A previous study analyzed an RNA-seq data performed in human subcutaneous and visceral adipose tissue, which indicated that visceral fat tissue *CDO1* mRNA was positively associated with important pathways for adipose tissue physiology, including fatty acid metabolism, PPAR signaling, thermogenesis, AMPK signaling, insulin signaling, mitochondrial activity and function[49]. Interestingly, according to our RNA-seq data, the AMPK signaling pathway was significantly enriched in downregulated genes of mice livers (Cdo1^LKO vs WT) (Fig. 3f). This reveals the possibility that Cdo1 participates in the regulation of AMPK signal pathway. Thus, hepatic Cdo1 could be an important downstream effector of exercise in the combat against NAFLD, in which AMPK signaling could be involved.

Our studies indicate that Cdo1^LKO mice had significant lower basal metabolic rates (Supplementary Fig. 2e-g), while Cdo1^LTG mice had significant higher basal metabolic rates (Supplementary Fig. 7g-i), which were compared to their corresponding control companions. Liver is known to be one of the high metabolic rate organs[50]. In our work, it was found that Cdo1 could activate AMPK signaling to enhance mitochondrial biogenesis and FAO in hepatocytes. The above pathways would promote hepatic catabolism to increase liver energy expenditure, which contributes to the increase of basal metabolic rates of the mice. In addition, Cdo1-induced AMPK signaling in hepatocytes may also trigger some cross-talks between the liver and other organs, such as fat tissue and muscle, through regulating the expression and secretion of some endocrine factors from hepatocytes, which could promote energy expenditure of the other organs to further enhance the metabolic rates of the mice. Thus, when Cdo1 was knocked out in hepatocytes, the above cell-autonomous and inter-organ communication processes might be blocked, which could impair the energy expenditure in the liver and even in some other organs, thereby decreasing the basal metabolic rates of the mice. Further studies are needed to corroborate above hypothesis.

Many studies have shown that the AMPK signaling plays an important role in the alleviation of hepatocyte steatosis by positively regulating lipid metabolism and mitochondrial function[44,45]. In our study, it was found that the knockdown of AMPK blunted the role of Cdo1 in improving hepatic steatosis (Fig. 6). These results confirm that AMPK signaling is critical for the protective role of Cdo1 against NAFLD. In the mechanistic study, we demonstrated that Cdo1 tethered Camkk2 to AMPKα, which promotes the AMPK activation to improve hepatocyte steatosis (Fig. 7 and Supplementary Fig. 6). The co-

immunoprecipitation (coIP) assays showed that Cdo1 could interact with both Camkk2 and AMPK simultaneously. When Cdo1 was overexpressed in primary hepatocytes, more Cdo1 could interact with Camkk2, which was accompanied with more AMPK binding to Camkk2 (Fig. 7e). Consistently, deficiency of Cdo1 in primary hepatocytes resulted in no Cdo1 interaction with Camkk2, which was accompanied with less AMPK binding to Camkk2 (Fig. 7g). The in vitro cell-free reconstitution experiment was also performed by using recombinant Cdo1, Camkk2 and AMPK proteins. In consistence with the cellular experiments that examined the role of Cdo1 in regulating AMPK kinase activity (Fig. 5c and Supplementary Fig. 4c), it was found that Cdo1 also promoted Camkk2-induced Thr172 phosphorylation of AMPKα in the in vitro cell-free kinase assay (Fig. 7i, j and Supplementary Fig. 6l). Together, these data demonstrate that Cdo1 can tether Camkk2 to AMPK by simultaneously binding with both of them, thereby facilitating the phosphorylation of AMPK by Camkk2. A scaffold is a protein that binds two or more proteins to promote reaction assembly and to increase the efficiency of a molecular event, such as signal transduction[51]. For example, Kinase Suppressor of Ras (KSR) functions as a scaffold to associate with both RAF and MEK. This will promote a closer contact of RAF with its substrate MEK to assemble the RAF/MEK functional pair, thereby facilitating the phosphorylation of MEK by RAF[52]. For another example, the scaffold protein NF-κB essential modulator (NEMO) can interact with both IKKβ and IκBα, which will direct the kinase activity of IKKβ towards IκBα to fuel the canonical NF-κB signaling[53]. Based on our CoIP experiments (Fig. 7a–h), cellular experiments for testing AMPK kinase activity (Fig. 5c and Supplementary Fig. 4c) and in vitro cell-free AMPK kinase assay (Fig. 7i, j and Supplementary Fig. 6l), it is suggested that Cdo1 could also function as a scaffold protein that links Camkk2 to AMPK by interacting with both of them, thereby positioning Camkk2 in close proximity to its substrate AMPK. This may increase local concentration of AMPK for Camkk2, promote the formation of Camkk2/AMPK functional pair and allow for more efficient Camkk2-mediated phosphorylation of AMPK. Furthermore, it is found that the catalytically inactive mutant form of Cdo1 (Y157F) is still capable of activating AMPK and eliciting protective effects in vitro. This may be because that this enzymatically inactive mutation does not affect the interaction of Cdo1 with Camkk2 and AMPK (Supplementary Fig. 6c). As a result, Cdo1 mutant (Y157F) could still act as a scaffold protein to promote the association between AMPK and Camkk2, which leads to the activation of AMPK signaling. Thus, these results above reveal a potential non-canonical function for Cdo1.

The mechanism of how exercise leads to the induction of hepatic Cdo1 was also investigated in our study. It was found that exercise induced the upregulation of cAMP level in mice livers (Fig. 1e). This was concomitant with the activation of CREB phosphorylation (Fig. 1c) that can be triggered by PKA kinase. We performed the ChIP assays to confirm the binding of CREB to the promoter of Cdo1 (Fig. 1f). We also constructed the Cdo1 promoter luciferase reporter gene and found that CREB could activate the transcription of Cdo1, in which one CREB binding site plays an important role in the transactivation (Fig. 1g–i). In consistence with the above results, upregulating the cAMP level by

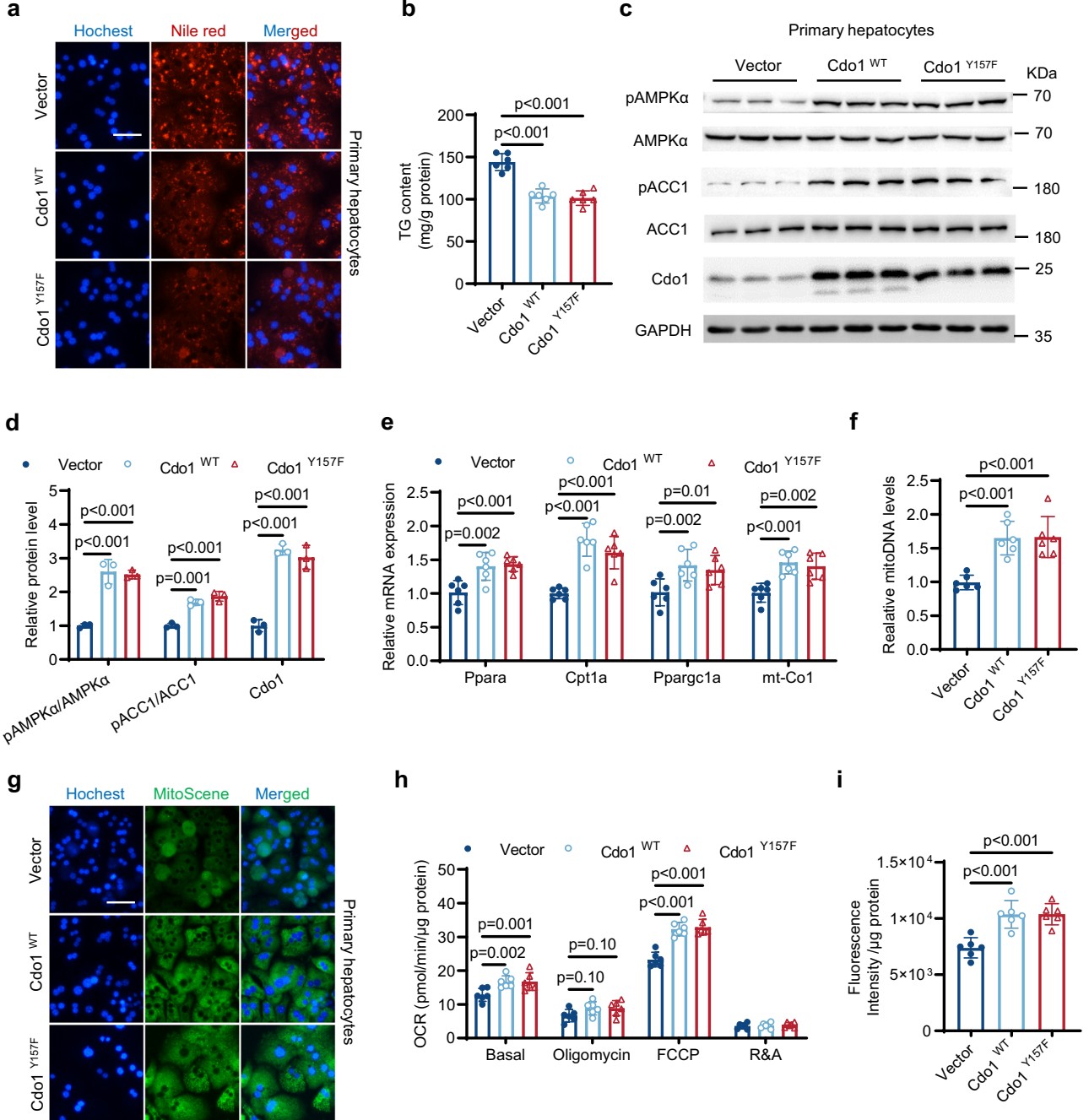

**Fig. 5 | Cdo1 overexpression alleviates hepatocytes steatosis in vitro.** WT primary hepatocytes were infected with adenovirus harboring Cdo1 (AD-Cdo1^WT), AD-Cdo1^Y157F or AD-LacZ (for Vector). After 24 h, cells were treated with a mixture of 0.6 mM FFAs (oleate and palmitate) at a final ratio of 2:1 for 24 h. Then cells were harvested and analyzed. $n = 3$ independent biological replicates for western blotting. $n = 6$ independent biological replicates for other experiments. **a** Nile red and Hoechst staining of the cells. Scale bars, 50 μm. **b** TG levels in primary hepatocytes. **c** Primary hepatocytes lysates were analyzed by western blotting with the indicated antibodies. GAPDH serves as an internal control. **d** Quantification of western blotting results of **c**. **e** The mRNA levels of the indicated genes were determined. **f** The mitochondrial DNA (mitoDNA) levels in primary hepatocytes. **g** MitoScene and Hoechst staining of the cells Scale bars, 50 μm. **h** Respiration in primary hepatocytes. OCR, oxygen consumption rate. FCCP, mitochondrial uncoupler. R&A, rotenone and antimycin A. **i** Fatty acid oxidation (FAO) levels in primary hepatocytes. Experiments were performed 3 times and similar results were obtained in **a**, **c** and **g**. One-way analysis of variance plus Tukey's post hoc tests were performed in **b**, **f** and **i**; two-way analysis of variance plus Tukey's post hoc tests were performed in **d**, **e** and **h**. Source data are provided as a Source Data file.

Forskolin promoted Cdo1 expression as well as CREB phosphorylation in hepatocytes (Fig. 1j-o). For one thing, hepatic Cdo1 can be induced in response to exercise through the cAMP/PKA/CREB, and functions as an exercise-response factor to play an important role in the alleviation of hepatosteatosis. For another thing, exercise positively promotes Cdo1 expression, and Cdo1-Camkk2-AMPK axis could be one molecular mechanism underlying the effect of exercise on the improvement in NAFLD, which may also provide new molecular evidence for prescribing exercise as a treatment for NAFLD.

There are some limitations in our work. First, we have confirmed that moderate-intensity continuous exercise can promote the expression of hepatic Cdo1 in mice. But the effects of different exercise patterns or amount of exercise on hepatic Cdo1 expression need to be further investigated. Second, because exercise can promote metabolic

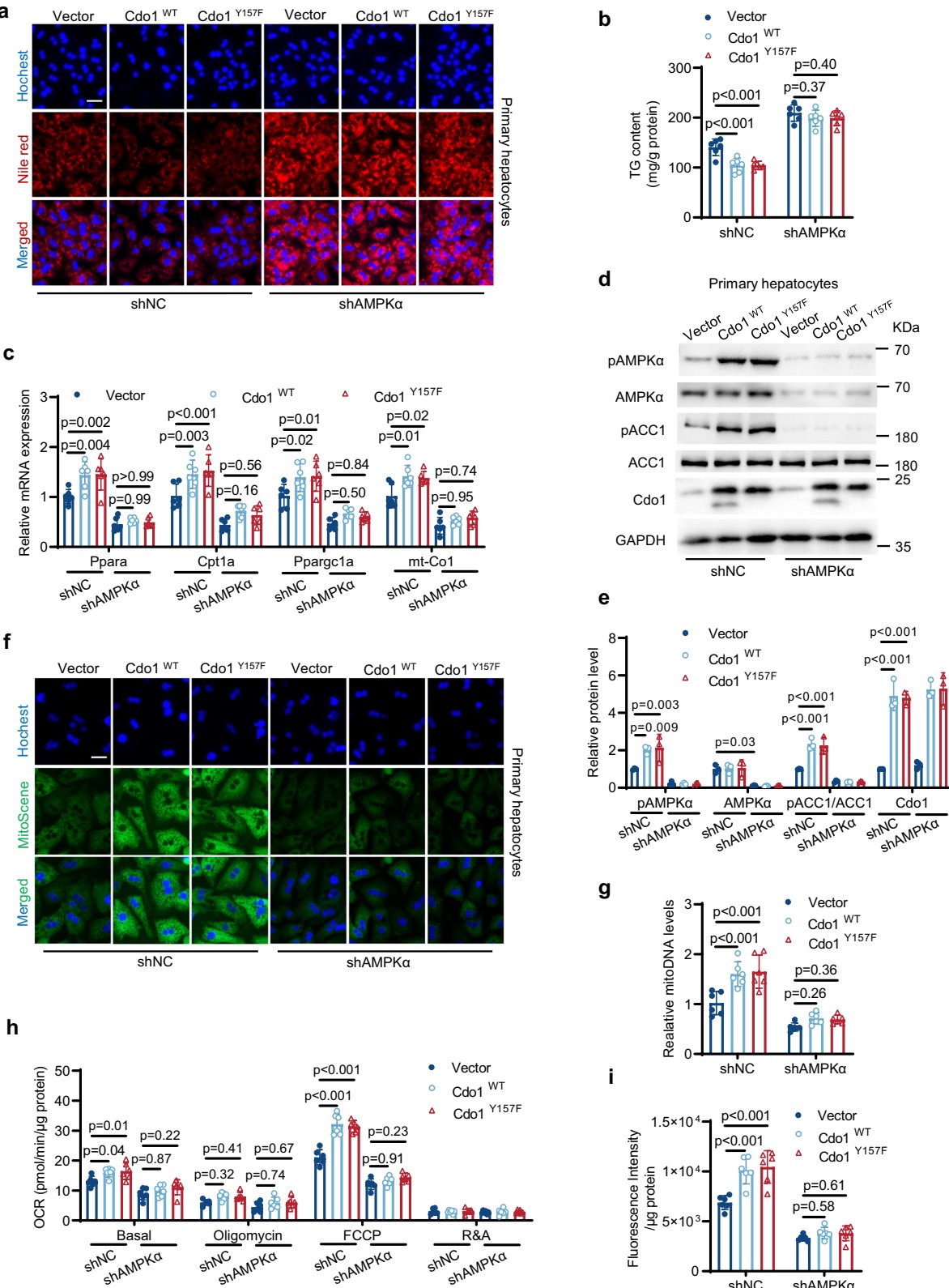

benefits, whether hepatic Cdo1 also plays roles in exercise-mediated metabolic improvement in other tissues needs to be further examined. Last, it has been mentioned in many cancer-related studies that Cdo1 expression is tightly controlled by promoter methylation, and hypermethylation of the Cdo1 promoter will lead to transcriptional inactivation and silencing of the Cdo1 gene[11,54,55]. Therefore, it remains to be investigated whether the decrease in hepatic Cdo1 expression is related to the increased methylation of Cdo1 promoter in NAFLD individuals.

In conclusion, our data demonstrate that exercise can promote the expression of hepatic Cdo1 through cAMP/PKA/CREB signaling pathway. Liver-derived Cdo1 plays an important role in exercise-mediated alleviation of hepatic steatosis in mice. Furthermore, hepatic Cdo1 and exercise can cooperate to alleviate fatty liver in a manner

**Fig. 6 | AMPK signaling is indispensable for Cdo1-mediated amelioration of hepatocytes steatosis.** Wild-type (WT) primary hepatocytes were infected with adenovirus harboring Cdo1 (AD-Cdo1$^{WT}$), AD-Cdo1$^{Y157T}$ or Vector, together with the infection of adenoviruses harboring the indicated short hairpin RNA (shRNAs). After 24 h, cells were treated with a mixture of 0.6 mM FFAs (oleate and palmitate) at a final ratio of 2:1 for 24 h. Then cells were harvested and analyzed. $n = 3$ independent biological replicates for western blotting. $n = 6$ independent biological replicates for other experiments. **a** Nile red and Hoechst staining of the cells. Scale bars, 50 μm. **b** Triglyceride (TG) levels in primary hepatocytes. **c** The mRNA levels of the indicated genes were determined. **d** Primary hepatocytes lysates were analyzed by western blotting with the indicated antibodies ($n = 3$ independent biological replicates, and representative blot was shown). GAPDH serves as an internal control. **e** Quantification of western blotting results of **d** was done by using ImageJ. **f** MitoScene and Hoechst staining of the cells. Scale bars, 25 μm. **g** The mitochondrial DNA (mitoDNA) levels in primary hepatocytes. **h** Respiration in primary hepatocytes. OCR, oxygen consumption rate. FCCP, mitochondrial uncoupler. R&A, rotenone and antimycin A. **i** Fatty acid oxidation (FAO) levels in primary hepatocytes. Experiments were performed 3 times and similar results were obtained in **a**, **d** and **f**. Two-way analysis of variance plus Tukey's post hoc tests were performed in **b**, **c**, **e**, **g**, **h** and **i**. All data show the means ± SD. Source data are provided as a Source Data file.

involving Cdo1-Camkk2-AMPK axis. Importantly, our findings provide evidence that hepatic Cdo1 could be an important downstream effector of exercise for the prevention and treatment of NAFLD.

## Methods

### Animals

All animal experiments were approved by Shanghai University of Sport Animal Care and Use Committee (Ethics No. 102772022DW002). All studies involving animal experimentation followed the National Institute of Health guidelines on the care and use of animals. Albumin-Cre mice were purchased from the Model Animal Research Center of Nanjing University. All male and female Cdo1$^{flox/flox}$ mice and Cdo1 transgenic mice were purchased from Shanghai Model Organisms Center. Cdo1$^{LKO}$ mice (Cdo1$^{flox/folx}$/Albumin-Cre$^+$) were generated by crossbreeding Cdo1$^{flox/flox}$ and albumin promoter driven Cre transgenic mice. Liver specific Cdo1 transgenic (Cdo1$^{LTG}$) mice were generated by cross-breeding the mice containing the CMV promoter-driven but stop signal-suppressed 3*Flag-tagged murine Cdo1 expression cassette (CMV-Stop-Cdo1 mice) with Albumin-Cre mice. Cdo1$^{LKO}$ mice and their wild type (WT) litter mates, and Cdo1$^{LTG}$ mice and their WT litter mates, were used in the experiments. C57BL/6 J mice were fed with high-fat diet (HFD, D12492, Research Diets) to induce hepatic steatosis, and the chow diet (CD, 1010086, Xietong Shengwu, China) was used as the control diet. All animals were housed at $23 \pm 2$ °C with a humidity of $50\% \pm 5\%$ in a 12 h light/dark cycle and fed ad libitum with standard mouse feed and water throughout the experiments. C57BL/6 J mice were used for all the experiments.

### Treadmill training and time to exhaustion

The treadmill exercise program includes 1 week of treadmill familiarization followed by 7 weeks (6 days per week) of treadmill training. During the treadmill training, we kept checking that all mice were running, but not resting, on the treadmill. This would guarantee that all mice were subjected to identical level of exercise training. The detailed treadmill program is illustrated in the corresponding figure. The time to exhaustion assay is described as follows. Before an exercise bout, chow diet-fed mice were placed in the treadmill (ZhengHuabiological, ZH-PT/5 S, China) for acclimation. Adaptive training was conducted on the first day (starting speed at 12 m/min for 10 min followed by 20 m/min for 20 min). The treadmill test was conducted on the second day. The starting speed was 5 m/min for 3 min, and the speed was increased by 5 m/min every 3 min until the speed reached 28 m/min. The maximum speed of 28 m/min was continued, and the time to exhaustion was measured.

### Mouse metabolic and liver function assays

For GTT, the mice were fasted for 18 h and received an intraperitoneal (i.p.) injection of D-glucose (2 mg/g body weight). For ITT, the mice were injected intraperitoneally with human insulin (Eli Lilly; 0.8 mU/g body weight) after 6 h fasting. Blood glucose levels were measured at 0, 15, 30, 60, 90 and 120 minutes after injection (a glucometer monitor, Roche). For the GTT/ITT assays, the area of the curve (AOC) was calculated using the conventional trapezoid rule. Levels of serum alanine aminotransferase (ALT), and aspartate aminotransferase (AST) were measured using the kits from Applygen, China (#E2021 and E2023, respectively) according to the manufacturer's instructions.

### Primary hepatocytes isolation and cells culture

Primary hepatocytes isolated from male Cdo1$^{flox/flox}$ or wild type mice aged 6 to 8 weeks by using the collagenase perfusion method. Briefly, after perfusion and digestion by collagenase IV solution (#C5138, Sigma), liver tissue was dissociated and filtered through a 70-μm cell strainer (#352350, Falcon). The resulting cell suspension was centrifuged at 50 g with 3 times to collect hepatocytes (cell pellets). Primary hepatocytes, HepG2, Hepa1-6 and HEK293T cells maintained in Dulbecco's Modified Eagle Media (DMEM) containing 10% fetal bovine serum (FBS, Gibco) at 37 °C in a 5% CO2 environment. The cell lines sources are as follows: HEK293T (American Type Culture Collection, CRL3216, USA), HEK293A (National Collection of Authenticated Cell Cultures, SCSP-5094, China), Hepa1-6 (American Type Culture Collection, CRL1830, USA), HepG2 (National Collection of Authenticated Cell Cultures, SCSP-510, China).

### Lipid analysis

To establish cellular models of lipid accumulation, primary hepatocytes were seeded in six-well plates. After cell adhesion, 0.6 mM a mixture of free fatty acids (FFAs) was added to the medium for 24 hours (at a final ratio of 2:1 with oleate and palmitate from Sangon Biotech, China). The cells were then fixed with 4% paraformaldehyde for 10 minutes and stained with nile red and Hoechst staining for 10 minute to visualize intracellular lipid accumulation. The triglyceride (TG), total cholesterol (TC) contents in isolated primary hepatocytes or liver tissue were measured using the kits from Applygen, China (#E1003 and E1005, respectively), according to the manufacturers' instructions.

### MitoScene analysis

When the cultured cells reach the appropriate density, the old medium was discarded and the preheated medium containing appropriate concentration of MitoScene Green (#M4064, US Everbright) was added, which was then incubated at 37 °C for 45 min. After that, the medium containing MitoScene Green was discarded and Hochest/PBS was added to the dish at 37 °C for 10 min. Cellular fluorescence were captured by Fluorescence Microscopy (Leica microscope).

### Cysteine and taurine levels analysis

The cysteine and taurine contents in liver tissue were measured using the kits from YITA China (#YT6089) and CIBO BIO, China (#CB11346-mu), according to the manufacturers' instructions.

### FAO level analysis

The cell culture medium was removed and the cells were washed with Hank's balanced salt solution (HBSS) buffer twice. FAOblue was dissolved in HBSS to the final concentration of 10 μM (#FDV-0033, Funakoshi). HBSS containing FAOblue was added to the cells at 37 °C for 45 min. The medium was then removed and the cells were washed

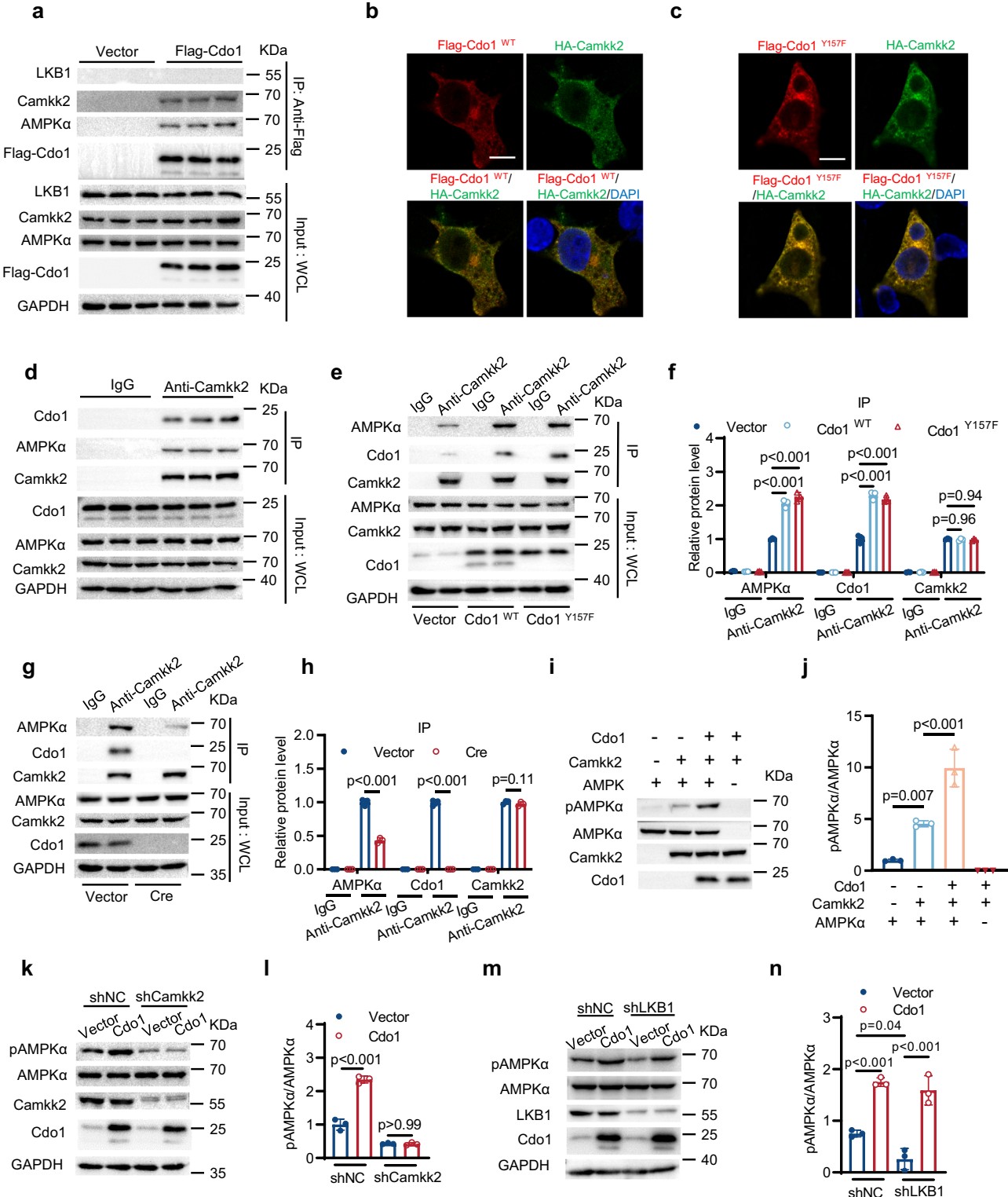

with HBSS twice. The cells were scraped off and the fluorescence intensity was detected by enzyme labeling instrument (detection wavelength: excitation of 405 nm/ emission of 460 nm, CLARIOstar by BMG LABTECH). The fluorescence intensity was compared to the protein concentration.

**Chromatin immunoprecipitation (ChIP)**

ChIP was performed as described previously[39]. Cells were cross-linked in 1% formaldehyde for 10 min at room temperature and then incubated with 125 mM glycine for termination. The cells were washed twice with ice-cold PBS, after centrifugation, resuspended in 500 ul of lysis buffer (50 mM Tris-HCl, pH 8.0, 1% SDS, 10 mM EDTA, and protease inhibitors). Samples were sonicated, and the average length of DNA fragments ranged between 200 and 800 bp. Samples were centrifuged at 12,000 g at 4 °C for 5 min. After removal of an input aliquot (whole-cell extract), supernatants were diluted 10-fold in ChIP dilution buffer (20 mM Tris-HCl, pH 8.0, 150 mM NaCl, 2 mM EDTA, 1% Triton X-100, and complete protease inhibitor tablets).

**Fig. 7 | Cdo1 tethers Camkk2 to AMPK by interacting with both of them, thereby promoting AMPK phosphorylation. a** Primary hepatocytes were infected with AD-Flag-Cdo1, and immunoprecipitation was performed using anti-Flag beads and followed by western blotting with antibodies as indicated ($n = 3$ independent biological replicates). WCL, whole cell lysate. **b, c** Hepa1-6 cells were co-transfected with plasmids encoding Flag-Cdo1$^{WT}$ or Flag-Cdo1$^{Y157F}$ and HA-Camkk2, and followed by confocal analyses. Scale bars, 10μm. **d** Cell lysates from primary hepatocytes was immunoprecipitated with antibody against Camkk2, followed by western blotting ($n = 3$ independent biological replicates). **e** Primary hepatocytes were infected with AD-Cdo1$^{WT}$ or AD-Cdo1$^{Y157F}$, and cell lysates were immunoprecipitated with antibody against Camkk2 and followed by western blotting ($n = 3$ independent biological replicates). **f** Quantification of immunoprecipitation (IP) results of **e. g** Cdo1$^{flox/flox}$ loci-containing primary hepatocytes were infected with AD-Cre for the ablation of Cdo1. Cell lysates were immunoprecipitated with antibody against Camkk2 and followed by western blotting ($n = 3$ independent biological replicates, and representative blot was shown). **h** Quantification of IP results of

**g. i** Cell free kinase assay with purified recombinant Cdo1, Camkk2 and AMPK proteins. The mixtures were detected by western blotting with antibodies as indicated ($n = 3$ independent biological replicates, and representative blot was shown). **j** Quantification of western blotting results of **i. k** Primary hepatocytes were infected with AD-Cdo1 and AD-shCamkk2 for 48 h. Cell lysates were then analyzed by western blotting ($n = 3$ independent biological replicates, and representative blot was shown). **l** Quantification of western blotting results of **k. m** Primary hepatocytes were infected with AD-Cdo1 and AD-shLKB1 for 48 h. Cell lysates were then analyzed by western blotting ($n = 3$ independent biological replicates, and representative blot was shown). **n** Quantification of western blotting results of **m.** Experiments were performed 3 times and similar results were obtained in **a–e, g, i, k** and **m.** One-way analysis of variance plus Tukey's post hoc tests was performed in **j;** two-way analysis of variance plus Tukey's post hoc tests were performed in **f, h, l** and **n.** All data show the means ± SD. Source data are provided as a Source Data file.

## Measurement of cellular oxygen consumption rate (OCR)
Mitochondrial respiration was measured by high-resolution respirometry (Oxygraph-2k, Oroboros, Innsbruck, Austria) at 37 °C. In brief, primary hepatocytes or HepG2 cells were suspended in 2 mL of DMEM in each chamber. Oligomycin, (1.5 μM, #S1478, Selleck), FCCP (1 μM, #S8276, Selleck), Rot (rotenone, 0.5 μM, #sc-203242, Santa Cruz) and antimycin A (0.5 μM, #sc-202467A, Santa Cruz) were added as indicated. After OCR measurement, the protein concentration of the cells were quantified.

## Luciferase reporter assays
Luciferase reporter assays was performed as described previously[56,57]. The proximal promoter regions of mouse Cdo1 and Cdo1 artificial mutants were subcloned into the firefly luciferase reporter construct PGL3-basic (Promega). HEK293T cells were transfected with 300 ng/well firefly luciferase reporter constructs and 6 ng/well Renilla luciferase reporter plasmids, in combination with 300 ng/well pcDNA3.1(-) vector or CREB plasmids, by using Lipofectamine 6000 (#C0526, Beyotime) according to the manufacturer's instructions. After 48 h, luciferase activity was measured using dual luciferase reporter assay (#RG027, Beyotime and CLARIOstar by BMG LABTECH), normalizing firefly luciferase to Renilla activity.

## RNA extraction and reverse transcription-quantitative PCR (RT-qPCR)
Total RNA from cells and tissues was extracted by using Trizol reagent (Invitrogen) according to the manufacturer's protocol. Complementary DNA (cDNA) was synthesized from total RNA by using reverse transcription kit (Beyotime, China). The cDNAs were amplified with Power SYBR green PCR master mix (Beyotime, China), with 18 S rRNA or 36b4 as an endogenous control. The results were collected from QuantStudio 6 Flex by Thermo Fisher Scientific instrument. The qPCR was done in triplicate and repeated at least 3 times. Primer information for RT-qPCR is listed in Supplementary information.

## Western blotting
Cells and tissues were harvested, prepared and western blotting was performed as described previously[58]. Lysates were run on SDS-PAGE gel (BIO-RAD) and subjected to western blotting with the primary antibodies to Cdo1 (Proteintech, Cat:12509-1-AP, Lot:00057877, dilution 1:1000), AMPKα (Proteintech, Cat: 10929-2-AP, Lot: 00115023, dilution 1:1000), p-AMPKα (Beyotime, Cat: AA393-1, dilution 1:1000), CREB1 (Proteintech, Cat: 12208-1-AP, Lot: 00102107, dilution 1:800), p-CREB (Proteintech, Cat: 28792-1-AP, Lot:00102484, dilution 1:800), ACC1 (Proteintech, Cat: 67373-1-IG, Lot: 10013129, dilution 1:1000), p-ACC1 (Santa Cruz, Cat: sc-271965, Lot: E1922, dilution 1:200), Camkk2 (Proteintech, Cat: 11549-1-AP, Lot: 00097881, dilution 1:800), p-Camkk2 (Affinity, Cat: AF4487, Lot: 8204594,dilution 1:500), LKB1(Santa Cruz, Cat: sc-32245, Lot: G1422,dilution 1:200), p-Akt(s473) (CST, Cat: 9272 S, Lot:28,dilution 1:1000), tAKT (CST, Cat: 4060 T, Lot:25, dilution 1:1000), HA (Proteintech, Cat: 51064-2-AP, Lot: 00116648,dilution 1:10000), Flag (Proteintech, Cat: 66008-4-Ig, Lot:10027647, dilution 1:6000), GAPDH (CST, Cat: D16H11, Lot: 8, dilution 1:1000), HSP90α/β (Santa Cruz, Cat: sc-13119, Lot: Jo722, dilution 1:1000); peroxidase affiniPure goat anti-mouse IgG secondary antibody (Jackon, Cat: 111-035-003, Lot: 151083, dilution 1:3000); peroxidase affiniPure goat anti-rabbit IgG secondary antibody (Jackon, Cat: 111-035-003, Lot: 153526, dilution 1:3000). Western blotting was developed and quantified by using Tanon-5200S (BIO-RAD) and Image J, respectively. And the values of target proteins were normalized to that of the internal control protein on the same membrane.

## Generation of recombinant adenoviruses and RNA interference
The recombinant adenoviruses and RNA interference were performed as described previously[39]. Recombinant adenoviruses (AD) for overexpression or knockdown were generated using ViraPower Adenoviral Expression System (Invitrogen). The targets (5′ to 3′) of short hairpin RNA (shRNA) harbored in the adenoviruses were listed as follows: non-specific control shRNA (shNC), CTACACAAATCAGCGATTT; shRNA against murine AMPKα1 (shAMPKα1), GCACACCCTGGATGAATTAAA; shRNA against murine AMPKα2 (shAMPKα2), GCTGAGAACCACTCCCTTTCT; shCamkk2, GTATCCACTTGGGCATGGAAT; shLKB1, GACAATATCTACAAGCTCTTT; shRNA against human AMPKα1, GTTGCCTACCATCTCATAATA; shRNA against human AMPKα2, GTGGCTTATCATCTTATCATT. Recombinant adenoviruses were produced and amplified in HEK293A cells. For adenovirus infection, primary hepatocytes or HepG2 cells were infected with the indicated adenoviruses and replaced with fresh DMEM after 24 h of viral infection. The cells were harvested for tests 48 h after infection. The small interfering RNAs (siRNAs) were designed and synthesized by Gene Pharma. The cells were harvested for the tests 48 h after transfection. Transfection was performed when cell growth reached 80% confluency and RNAiMAX

Then, the samples were divided equally, and 10% of each sample was used for input control. The samples were precleared using ChIP Grade Protein A/G magnetic beads (Thermo Fisher, 26162) for 1 h at 4°C and immunoprecipitated with the indicated antibodies of anti-CREB (Proteintech, 2208-1-AP), and control IgG (Abcam, ab46540). Immunoprecipitated samples were eluted and reverse cross-linked by incubation overnight at 65 °C in elution buffer (50 mM Tris-HCl, pH 8.0, 10 mM EDTA, 1% SDS). Genomic DNA was then extracted with a PCR purification kit (Qiagen). Purified DNA was subjected to qPCR using primers specific to the promoters of the indicated genes. The primers for ChIP-qPCR are listed in Supplementary information.

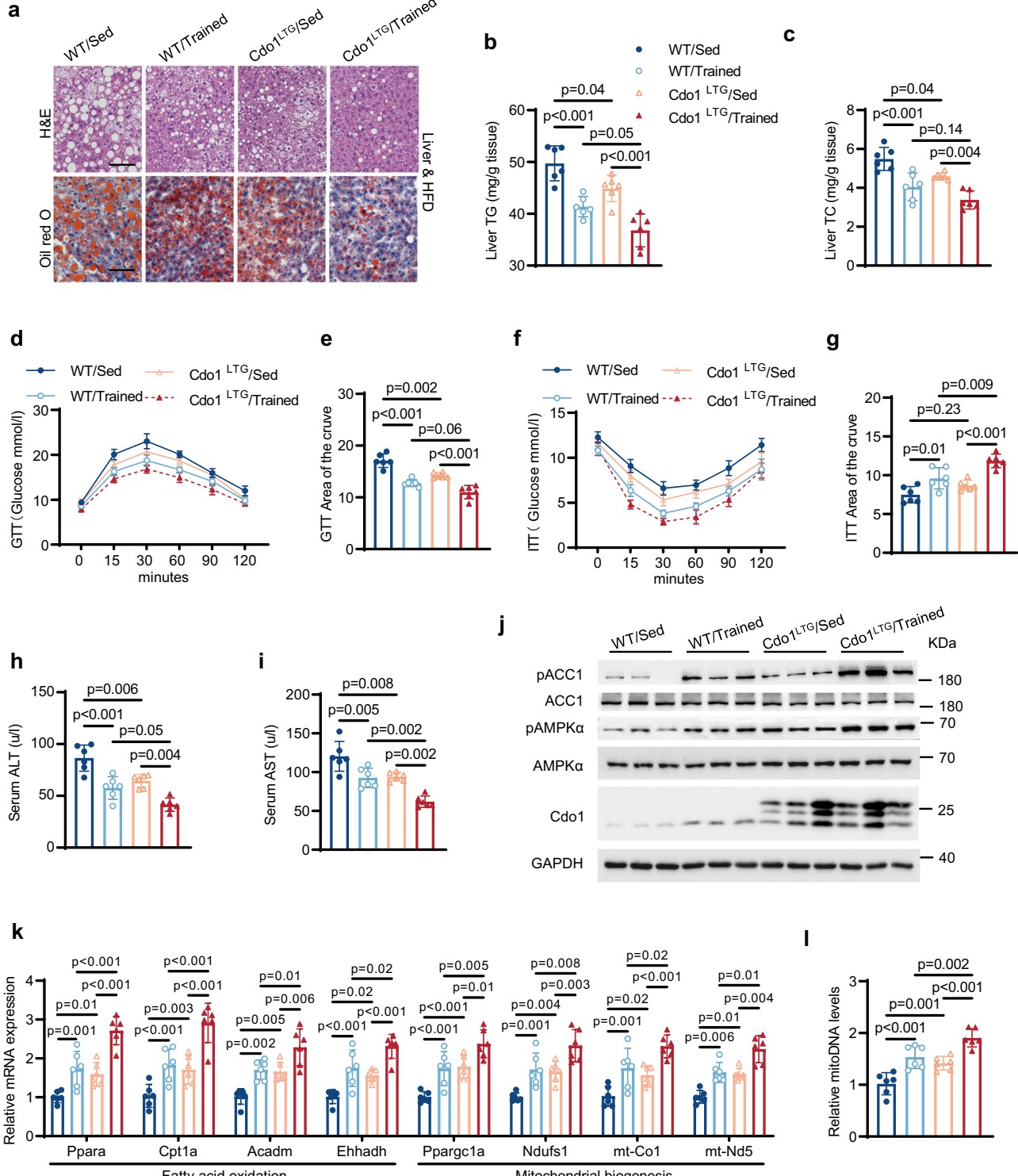

**Fig. 8 | Hepatocyte-specific overexpression of Cdo1 (Cdo1^LTG) and exercise synergistically alleviate NAFLD in mice.** WT and Cdo1^LTG male mice under high-fat diet (HFD) feeding 16 weeks, with or without exercise for the last 8 weeks. Then mice were sacrificed for analysis. **a** Representative images of hematoxylin and eosin (H&E) staining and Oil Red O staining. Experiments were performed 3 times and similar results were obtained. Scale bars, 50 μm. **b, c** Triglyceride (TG) and cholesterol (TC) levels in mice livers ($n = 6$ mice per group). **d** Glucose tolerance test (GTT) was performed in mice under 14 weeks of HFD feeding ($n = 6$ mice per group). **e** Analysis of the GTT data in **d**, with subtraction of the basal glucose to generate an area of the curve (AOC). **f** Insulin tolerance test (ITT) was performed in mice fed with 15 weeks of HFD ($n = 6$ mice per group). **g** Analysis of the ITT data in **f** with AOC. **h, i** Serum alanine aminotransferase (ALT) and serum aspartate aminotransferase (AST) levels in mice, respectively ($n = 6$ mice per group). **j** The mice liver lysates were analyzed by western blotting ($n = 3$ mice per group). **k** The mRNA levels of the indicated genes ($n = 6$ mice per group). **l** The mitochondrial DNA (mitoDNA) levels in mice livers ($n = 6$ mice per group). Experiments were performed 3 times and similar results were obtained in **a** and **j**. Two-way analysis of variance plus Tukey's post hoc tests were performed in **b, c, e, g, h, i, k** and **l**. All data show the means ± SD. Source data are provided as a Source Data file.

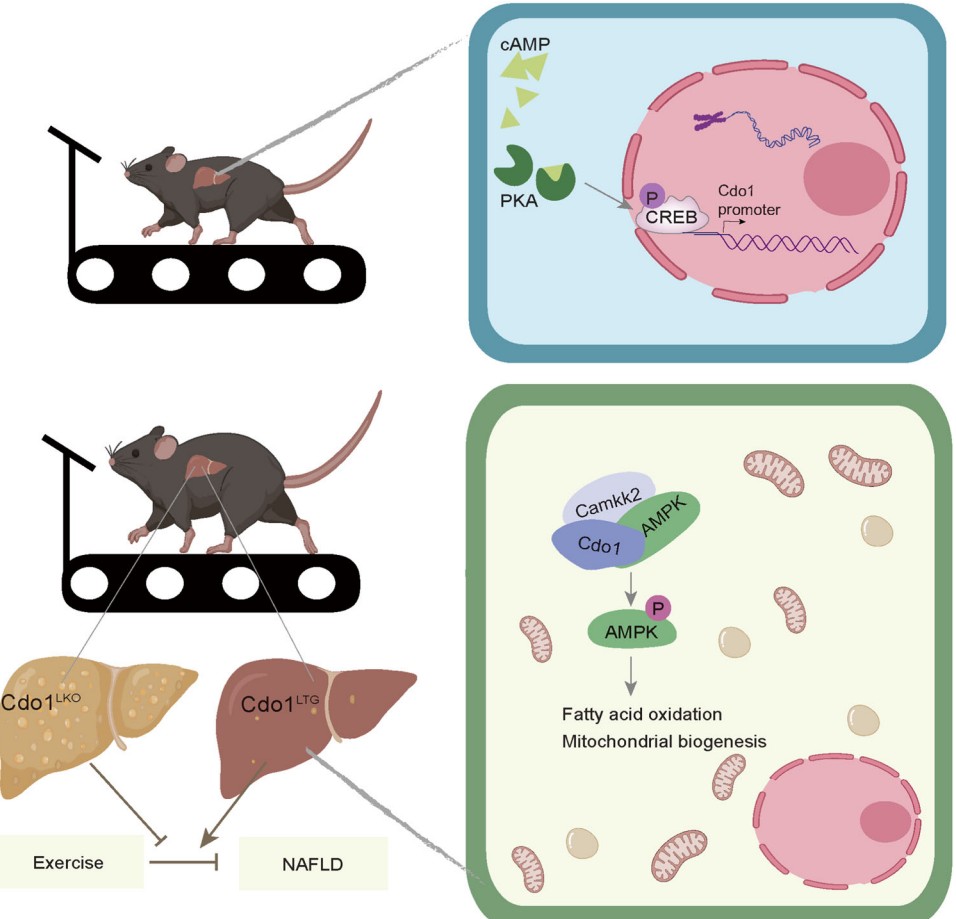

**Fig. 9 | A graph model for the role of hepatic Cdo1 in exercise-mediated protective effects against NAFLD.** Exercise can promote the expression of Cdo1 through cAMP/PKA/CREB signaling pathway. CREB can bind to Cdo1 promoter and promote Cdo1 transcription. Cdo1$^{LKO}$ impairs exercise-mediated effects against NAFLD in mice. On the contrary, Cdo1$^{LTG}$ and exercise can cooperate to alleviate hepatocyte steatosis. Mechanistically, Cdo1 tethers Camkk2 to AMPK by interacting with both of them, thereby promoting AMPK phosphorylation. This enhances fatty acid oxidation and mitochondrial biogenesis and would promote cellular energy expenditure, thereby inhibiting hepatosteatosis. Hepatic Cdo1 could function as an exercise-responsive effector in fighting against NAFLD. The representing mouse, mitochondria and liver images in panel were created using BioRender.com.

(#13778-100, Invitrogen) was used for the transfection according to the manufacturer's protocol. The nonspecific siRNA control (siNC) was used as a negative control. The target sequences for successful siRNAs (5′ to 3′) were as follows: siCDO1-1, CAGUACAGGUAUACC CGAATT; siCDO1-2, GGGAAAUCUAAAGGAGACATT; siNC, TTCTCCG AACGTGTCACGT. The siCDO1-1 and siCDO1-2 target different sites of the human gene of CDO1.

### H&E staining, Oil O staining and immunofluorescence assay
The tissues were fixed in 4% paraformaldehyde and sectioned after being paraffin embedded. H&E and Oil O staining was performed as described previously (Olympus microscope)[58]. For immunofluorescence, HEK293T cells and Hepa1-6 cells transfected with plasmid (Flag-Cdo1$^{WT}$, Flag Cdo1$^{Y157F}$, HA-Camkk2 or HA-LKB1) were first fixed in 4% paraformaldehyde for 10 min, rinsed with PBS for three times, and then permeabilized using 0.2% Triton X-100 for 5 min. The cells were rinsed with PBS again, blocked in 2% BAS/PBS for 1 h, and incubated with rabbit HA (Proteintech, Cat: 51064-2-AP, Lot: 00116648, dilution 1:100), Flag (Proteintech, Cat: 66008-4-Ig, Lot:10027647, dilution 1:100) for 3 h in room temperature. They were washed with PBS and incubated with Goat anti-mus IgG/Alex Fluor 555 (Beyotime, Cat: A0460, dilution 1:500) and Goat anti-Ribbit IgG/Alex Fluor 488 (Beyotime, Cat: A0562, dilution 1:500) for 1 h. Then the cells were rinsed with PBS three times and mounted using an anti-fade mounting

medium (with DAPI) (#S2110, Solarbio). The immunofluorescence-stained cells were observed under a Leica microscope.

### The mitoDNA content quantification by quantitative real-time PCR
Mice tissues or cells were homogenized and digested with proteinase K overnight in a lysis buffer for DNA extraction by using the DNease blood and tissue kit (Qiagen) following the manufacturer's instructions. The results were collected from QuantStudio 6 Flex by Thermo Fisher Scientific and calculated from the difference in the threshold cycle ($\triangle$CT) values for mitoDNA and nuclear-specific gene via the amplification method of quantitative real-time PCR. The data are expressed as mitoDNA-CO1 gene normalized to nuclear-specific gene hexokinase-2 as described previously[59]. The primers used are listed in Supplementary Table 1.

### Detection of mice metabolic rate
Six-week-old mice were fed with HFD for 6 weeks. Energy expenditure was assessed using indirect calorimetry (TSE-system, XYZ, 6 M/R, Germany Instruments). The concentrations of oxygen and carbon dioxide were monitored at the inlet and outlet of the sealed chambers to calculate oxygen consumption and carbon dioxide production. Food intake and physical activity was also measured during these processed. Each chamber was measured at an interval of 1 h.

## Cell-free kinase assay

AMPK recombinant protein complex 200 ng (#CT005-H0907B, Sino-Biological) was incubated with or without recombinant Camkk2 300 ng (#PR7777A, Thermofisher) and Cdo1 125 ng (#HY-P70066, MedChemExpress) in a kinase buffer (50 mM Tris, pH 7.6, 0.5 mM DTT, 10 mM MgCl2, 200 μM ATP, 200 μM AMP, 0.5 mg/ml BSA, 1 mM CaCl2, 100 ng calmodulin). The above reaction mixture was incubated for 30 min at 30 °C with a total volume of 50 μl. Then AMPKα phosphorylation at Thr172 was detected by western blotting by using (p)-AMPKα (Thr172) antibody.

## RNA-seq analysis

Total RNA was obtained from the liver of HFD-fed control mice and Cdo1$^{LKO}$ mice. The transcriptome sequencing experiments were performed by Novogene Company (Beijing, China). The RNA-seq was performed in IIlumina Hiseq Platform and clean reads were aligned to the reference genome using Hisat2 v2.0.5. The differential expression analysis (Fig.3a) was performed using the DESeq2 R package (1.20.0). GO enrichment (Fig.3b) and KEGG analysis (Fig.3f and Supplementary Fig. 2o) of differentially expressed genes was implemented by the clusterProfiler R package (3.8.1), in which gene length bias was corrected. KEGG and GO terms with corrected p-value less than 0.05 were considered significantly enriched by differential expressed genes.

## Statistical analyses

All data showed the means ± standard deviation (SD) of at least three biological replicates with the n indicated in each experiment. The statistical analyses were indicated in the legends of each figure, with $p < 0.05$ indicating a statistically significant difference. The statistical analysis was performed in Graphpad prism (Version 9) and IBM SPSS Statistics 27.

## Reporting summary

Further information on research design is available in the Nature Portfolio Reporting Summary linked to this article.

## Data availability

The raw RNA-seq data generated in Fig. 3a, b, f and Supplementary Fig. 2o have been deposited in the GEO datasets under accession code GSE239729. The published microarray data generated from the liver tissue of obese mice (GSE83596) and RNA-seq data from the livers of patients (GSE126848) with or without NAFLD in the GEO database were analyzed and shown in Fig. 2a and b, respectively. The images representing mouse models, mitochondria and liver in Figs. 1a, 2d, 9 and Supplementary Fig. 7b are created with BioRender.com (agreement number: EJ264NJ98Y). The data used to generate the main results shown in the main figures and Supplementary figures are available as Source data. Source data are provided with this paper.

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

## Acknowledgements

This work was supported by The Program for Overseas High-level talents at Shanghai Institutions of Higher Learning (TP2022100 to L.G.), the National Natural Science Foundation of China (No. 32070751 and 31871435 to L.G), and was partially supported by a fund of Peak Disciplines (Type IV) of Institutions of Higher Learning in Shanghai.

## Author contributions

M.C. was involved in study design, conducted the experiments, analyzed the data and drafted the paper. J.-Y.Z., W.-J.M., H.-Y.L., Y.L., S.L., L.-J.Y. and R.-Y.L. performed the experiments. L.G. conceived the idea, designed and supervised the study, obtained the funding and cowrote the paper.

## Competing interests

The authors declare no competing interests.
