## [Peer Review File · Nature Communications]

Cdo1-Camkk2-AMPK axis confers the protective effects of exercise against NAFLD in miceEditorial Note: This manuscript has been previously reviewed at another journal that is not operating a transparent peer review scheme. This document only contains reviewer comments and rebuttal letters for versions considered at *Nature Communications*.

REVIEWER COMMENTS

Reviewer #1 (Remarks to the Author):

The revised version of this manuscript is improved. I still have some comments:

The reviewers have misunderstood my comment. I was asking for proof of physical training not effects of genotype as you can see below:

Question #5: As this is an exercise training paper, it would be expected to show some sort of traditional training effects like maximal running speed, mitochondrial adaptations or the like. Answer: According to the reviewer's suggestion, incremental-load exhaustive exercise was tested in the mice on treadmill. As shown in Supplementary Fig. 2b and Supplementary Fig. 7c, the running time to exhaustion was similar between WT and Cdo1 KO mice, and between WT and Cdo1 transgenic mice. These results indicate that the exercise capacity of mice was not affected by hepatic knockout or overexpression of Cdo1.

Regarding the metabolic chamber measurements they strongly support that basal metabolic rate is decreased by the KO but stats are borderline. This is likely due to the rather low number of observations. Since this is an important piece of information I suggest the authors perform another round of experiments to get proper stats and if the conclusion is that metabolic rate is affected by KO then this should be discussed more clearly in the manuscript and in the abstract.

Mistake line 151 fig S1 should be S2

Reviewer #2 (Remarks to the Author):

The authors have adequately addressed my concerns. Thank you.

Reviewer #3 (Remarks to the Author):

Reviewer 1's Q2 additional question

Data or consideration on how CDO enhances the AMPK phosphorylation ability of CAMKK2 (not simple binding) is needed.

Reponses to reviewer #1:

Question #1:

The reviewers have misunderstood my comment. I was asking for proof of physical training not effects of genotype as you can see below:

Question #5: As this is an exercise training paper, it would is expected to show some sort of traditional training effects like maximal running speed, mitochondrial adaptations or the like. Answer: According to the reviewer's suggestion, incremental-load exhaustive exercise was tested in the mice on treadmill. As shown in Supplementary Fig. 2b and Supplementary Fig. 7c, the running time to exhaustion was similar between WT and Cdo1 KO mice, and between WT and Cdo1 transgenic mice. These results indicate that the exercise capacity of mice was not affected by hepatic knockout or overexpression of Cdo1.

Answer:

We apologize for misunderstanding the reviewer's comment. In the revised manuscript, the effect of exercise training on the exercise capacity of mice was examined. Incremental-load exhaustive exercise was tested in the mice on treadmill. 8 weeks of exercise training led to significant increase of the running time to exhaustion in both WT and Cdo1^{LKO} mice (**Supplementary Fig. 2b**), and also significantly increased the running time to exhaustion in both WT and Cdo1^{LTG} mice (**Supplementary Fig. 7c**). These data indicate that exercise training did enhance the exercise capacity of the mice. Please see these changes that are highlighted in yellow in the section of **Results** in the revised manuscript.

We hope that we have got the point and met the requirement of the reviewer. If we still misunderstand the reviewer's comment, please let us know. We will continue to revise it.

Question #2:

Regarding the metabolic chamber measurements they strongly support that basal metabolic rate is decreased by the KO but stats are borderline. This is likely due to the rather low number of observations. Since this is an important piece of information I suggest the authors perform another round of experiments to get proper stats and if the conclusion is that metabolic rate is affected by KO then this should be discussed more clearly in the manuscript and in the abstract.

Answer:

We appreciate the reviewer's suggestion, which helps to make our data and results more rigorous and conclusive. In the revised manuscript, the data for metabolic chamber measurements were analyzed by using the more rigorous regression-based analysis with analysis-of-covariance (ANCOVA) method utilizing body mass as covariate, instead of using the methodology of mass-adjusted metabolic rate indices. The method of regression-based analysis with ANCOVA is recommended by the journal *Nature Metabolism* (PMID: 34489606), and helps to maximize the reliability and conclusiveness of the metabolic rate data.

By using the new statistical method to analyze our metabolic rate data, it was found

that hepatic knockout or overexpression of Cdo1 did significantly affect the basal metabolic rate of the mice. Compared to their WT companions, Cdo1 KO mice had significant lower O₂ consumption rates, CO₂ emission levels and heat production (**Supplementary Fig. 2e-g**), while Cdo1 OE mice had significant higher the above parameters (**Supplementary Fig. 7g-i**). However, neither Cdo1 KO nor Cdo1 OE affected food intake and physical activity of the mice (**Supplementary Fig. 2h, i** and **Supplementary Fig. 7j, k**). Therefore, altered metabolic rate in mice may contribute to the changes in body weight which is mediated by hepatic knockout or overexpression of Cdo1.

The important information that hepatic knockout or overexpression of Cdo1 significantly affected the basal metabolic rate of the mice has been included in the **Abstract** of the revised manuscript (highlighted in yellow). In the revised **Abstract**, it is mentioned that “Hepatocyte-specific knockout of Cdo1 (Cdo1^{LKO}) decreases basal metabolic rate of the mice and impairs the effect of exercise against NAFLD, whereas hepatocyte-specific overexpression of Cdo1 (Cdo1^{LTG}) increases basal metabolic rate of the mice and synergizes with exercise to ameliorate NAFLD”.

The above important information has also been discussed in more detail in the section of **Discussion** (highlighted in yellow), which is also shown as following:

Our studies indicate that Cdo1^{LKO} mice had significant lower basal metabolic rates (**Supplementary Fig. 2e-g**), while Cdo1^{LTG} mice had significant higher basal metabolic rates (**Supplementary Fig. 7g-i**), which were compared to their corresponding control companions. Liver is known to be one of the high metabolic rate organs (PMID: 35358549). In our work, it was found that Cdo1 could activate AMPK signaling to enhance mitochondrial biogenesis and FAO in hepatocytes. The above pathways would promote hepatic catabolism to increase liver energy expenditure, which contributes to the increase of basal metabolic rates of the mice. In addition, Cdo1-induced AMPK signaling in hepatocytes may also trigger some cross-talks between liver and other organs, such as fat tissue and muscle, through regulating the expression and secretion of some endocrine factors from hepatocytes, which could promote energy expenditure of the other organs to further enhance the metabolic rates of the mice. Thus, when Cdo1 was knocked out in hepatocytes, the above cell-autonomous and inter-organ communication processes might be blocked, which could impair the energy expenditure in liver and even in some other organs, thereby decreasing the basal metabolic rates of the mice. Further studies are needed to corroborate above hypothesis.

Please see these changes that are highlighted in yellow in the revised manuscript. We hope that we could address this issue raised by the reviewer.

Question #3:

Mistake line 151 fig S1 should be S2

Answer:

We thank the reviewer for pointing out our mistake, which has been corrected in the revised manuscript. Please see this change that is highlighted in yellow in the section of **Results** in the revised manuscript.

Response to reviewer #2:

Comment:

The authors have adequately addressed my concerns. Thank you.

Reply:

We thank the reviewer's valuable and constructive comments and suggestions, which have greatly improved our manuscript.

Response to reviewer #3:

Question #1:

Reviewer 1's Q2 additional question

Data or consideration on how CDO enhances the AMPK phosphorylation ability of CAMKK2 (not simple binding) is needed.

Answer:

We appreciate the reviewer's comment. This issue has been discussed in more detail in the section of **Discussion** (highlighted in yellow), which is also shown as following:

A scaffold is a protein that binds two or more proteins to promote reaction assembly and to increase the efficiency of a molecular event, such as signal transduction (PMID: 36273588). For example, Kinase Suppressor of Ras (KSR) functions as a scaffold to associate with both RAF and MEK. This will promote a closer contact of RAF with its substrate MEK to assemble the RAF/MEK functional pair, thereby facilitating the phosphorylation of MEK by RAF (PMID: 11850406). For another example, the scaffold protein NF- κ B essential modulator (NEMO) can interact with both IKK β and I κ B α , which will direct the kinase activity of IKK β towards I κ B α to fuel the canonical NF- κ B signaling (PMID: 22633953). Based on our CoIP experiments (**Fig. 7a-h**), cellular experiments for testing AMPK kinase activity (**Fig. 5c and Supplementary Fig. 4c**) and in vitro cell free AMPK kinase assay (**Fig. 7i, j and Supplementary Fig. 6l**), it is suggested that Cdo1 could also function as a scaffold protein that links Camkk2 to AMPK by interacting with both of them, thereby positioning Camkk2 in close proximity to its substrate AMPK. This may increase local concentration of AMPK for Camkk2, promote the formation of Camkk2/AMPK functional pair and allow for more efficient Camkk2-mediated phosphorylation of AMPK.

Therefore, it is proposed that the role of Cdo1 in enhancing the AMPK phosphorylation ability of Camkk2 may be because of its potential scaffold function. Cdo1 could act as a scaffold protein to bind with both AMPK and Camkk2, which would tether the two molecules and increase the chance of closer contact between AMPK and Camkk2. The above process could help to direct the kinase activity of Camkk2 to AMPK better and enhance the efficiency of Camkk2-mediated phosphorylation of AMPK.

We hope that we could address the reviewer's concern on this issue.

REVIEWERS' COMMENTS

Reviewer #1 (Remarks to the Author):

The authors have responded nicely to my final requests. No further comments.

Reviewer #3 (Remarks to the Author):

The authors have adequately addressed my concerns. Thank you.